# Self-recovering passive cooling utilizing endothermic reaction of $NH_4NO_3/H_2O$ driven by water sorption for photovoltaic cell

Seonggon Kim[1], Jong Ha Park[2], Jae Won Lee[3], Yongchan Kim[1,4] & Yong Tae Kang ◎ [1,4] ✉

Power efficiency of photovoltaic cell is significantly affected by the cell temperature. Here, a self-recovering passive cooling unit is developed. The water-saturated zeolite 13X is coated on the back side of photovoltaic cell, and ammonium nitrate is dispersed as a layer to form a thin film. When heat is supplied, water is desorbed from zeolite 13X (latent cooling), and dissolves ammonium nitrate to induce endothermic reaction cooling. It is a reversible process that recovers itself at night. The unit works on the basis that the water sorption performance of porous materials is inversely proportional to temperature, and the solubility of endothermic reaction pairs increases proportionally with temperature. The average temperature of photovoltaic cell can be reduced by 15.1 °C, and the cooling energy density reaches 2,876 kJ/kg with average cooling power of 403 W/m². We show that highly efficient passive cooling comprising inexpensive materials for photovoltaic cell could be achieved.

Solar energy is widely utilized to reduce the grid's energy load[1]. Photovoltaic (PV) cells convert energy from the sun directly to electricity, and are simple to set up because they have few components. The efficiency of a PV cell is approximately 10–25% considering that it can solely utilize a specific range of wavelengths. To date, although highly efficient PV cells have been developed including the perovskite cell[2], building-integrated PV modules are mostly composed of monocrystalline silicon with high technical maturity[3]. On the other hand, the power efficiency of PV devices is significantly affected by the cell temperature[4]. For instance, the power efficiency of monocrystalline PV is reduced by 33.3% when the junction temperature is increased from 20 °C to 60 °C[5]. A greater reduction in efficiency by the temperature increase has also been reported for the perovskite cells[6]. During electricity generation, the temperature rise owing to solar radiation is inevitable, and various cooling devices have been developed for stable power generation including passive cooling and active cooling strategies[7,8].

Passive cooling methods generate cooling effect permanently without energy consumption[9]: Natural air cooling is a phenomenon in which convective heat transfer occurs, owing to the density difference when the temperature of air rises around the cell. Elminshawy et al.[10] reduced the average working temperature of PV cell by 19% utilizing the natural air cooling with attached fins at a submerged area ratio of 20%. Natural liquid cooling composed of PV modules and a fluid storage tank was studied[11]. The cooling effect of natural liquid cooling can be improved by using nanofluid as a coolant in microchannels[12]. The phase change cooling technologies have been actively studied utilizing two-phase coolant with boiling temperature of 34 °C[13], RT35[14], $Al_2O_3$ composites[15], and paraffin-based materials[16]. The heat pipe cooling device has been proposed and a maximum thermal efficiency can be improved by 14.2%[17]. Li et al.[18] developed atmospheric water sorption-based cooling for PV cell, which can also harvest water during power generation. The radiant cooling methods have been also discussed[19]. Li et al. studied the nighttime radiative cooling for PV cell with water

[1]Research Center for Plus Energy Building Innovative Technology, 145 Anam-ro, Seongbuk-gu, Seoul 02841, Republic of Korea. [2]Department of Mechanical Engineering, University of California, Berkely, Berkely, CA 94720, USA. [3]Division of Mechanical Engineering, Korea Maritime & Ocean University, 727 Taejong-ro, Yeongdo-gu, Busan 49112, Republic of Korea. [4]School of Mechanical Engineering, Korea University, 145 Anam-ro, Seongbuk-gu, Seoul 02841, Republic of Korea. ✉e-mail: ytkang@korea.ac.kr

harvesting[20]. Recently, co-generation of electricity and freshwater[21], dynamic photovoltaic building envelopes[22], and PV cooling-driven seawater desalination[23] were discussed as the PV cooling strategies.

Active cooling methods consume energy to achieve a cooling effect: Forced air cooling generates air velocity through a fan, which has a better cooling effect than the natural convection[24]. However, when the ambient temperature is high, the cooling effect significantly decreases, and the maintenance cost becomes high as well[9]. In addition, other challenges, such as space efficiency and energy consumption, make it less applicable. Hydraulic and thermoelectric cooling techniques, which, respectively, adopt working fluids and electricity as coolants, are not economically feasible[25]. On the other hand, PV thermal systems (PVTs) have been developed. As the fluid circulates at the bottom of PV cell through the channels, the thermal energy is harvested and PV cell cools down. PVTs are feasible in that the harvested thermal energy can be utilized in buildings, although the energy is consumed to circulate fluid. The $Al_2O_3$ nanofluid[26], $SiO_2$ nanofluid[27], and microencapsulated PCM slurry fluid[28] were discussed as working fluids. Weinstein et al.[29] proposed beam splitting-based hybrid electric and thermal solar receiver. To summarize, while low-cost cooling methods exhibit a low cooling effect, the methods with high cooling efficiencies are expensive at the present level of technologies.

Here, we demonstrate a passive cooling unit for a PV cell to realize the best performance among the cooling methods that have been reported in the literature[9,30]. The cooling system induced by a chain reaction of latent cooling (water desorption process) and endothermic reaction (dissolution process) has not been attempted for PV applications, which is the working principle of this study. The proposed method has no energy consumption and consists of cheap materials, making it highly applicable. Moreover, it can be repeatedly utilized, owing to its self-recovering characteristics. Meanwhile, a similar thermal management strategy for electronic devices was reported[31]. As heat is generated in electronic devices, the water molecules trapped in metal-organic frameworks (MOFs) are desorbed, which produces latent cooling effect. In this study, we have improved the atmospheric water desorption-driven cooling system by the chain reaction in which the desorbed water melts the crystal layer of ammonium nitrate and the secondary endothermic reaction cooling in a closed thin film.

## Results

### Working principle of self-recovering passive cooling

The working principle for the self-recovering passive cooling unit is presented in Fig. 1. A passive cooling unit is attached to the backside of the PV cell, and the power terminal unit is waterproofed so that it is not affected by moisture as presented in Fig. 1a. The passive cooling unit is driven by a chain reaction: when heat is supplied from the outside (solar radiation), solvent-saturated porous materials absorb heat, and desorb the solvent (primary latent cooling). The regenerated solvent generates secondary cold energy by dissolving the solute (endothermic reaction cooling). Therefore, the presented cooling method is defined as a water desorption-driven endothermic reaction (WD-ER) cooling unit. The theoretical cooling power of the PV cell with WD-ER is presented in Fig. 1b. Initially, the temperature rises owing to solar radiation. Subsequently, the temperature remains approximately constant as heat is absorbed into the porous materials, which is analogous to the PCM cooling method[32]. Solute is dissolved in a solvent with endothermic heat and actively decreases the temperature. These phenomena occur repeatedly, corresponding to the exterior heat, and the power efficiency of the PV cell will increase as well.

The operational mechanisms of WD-ER unit are discussed in Figs. 1c, d for the optimal compositions. The cooling effect can be obtained because the water adsorption capacity of zeolite 13X is inversely proportional to the temperature[33], and water can dissolve more ammonium nitrate at higher temperatures[34]. At an ambient temperature of 20−30 °C, the water is saturated with zeolite 13X, which acts as a solvent carrier, and ammonium nitrate is in a solid state without water. As heat is supplied, the water adsorption capacity of zeolite 13X decreases when desorption occurs, and the ammonium nitrate is dissolved in water as the solubility increases. During power generation, the heat of 27.4 kJ/mol-$H_2O$ is absorbed by zeolite 13X coated at the back of the PV cell. Liquid $H_2O$ then dissolves $NH_4NO_3$ into $NH_4^+$ and $NO_3^-$, which induce a heat absorption of 28.1 kJ/mol-$NH_4NO_3$. At night, when the temperature is low, the ammonium nitrate

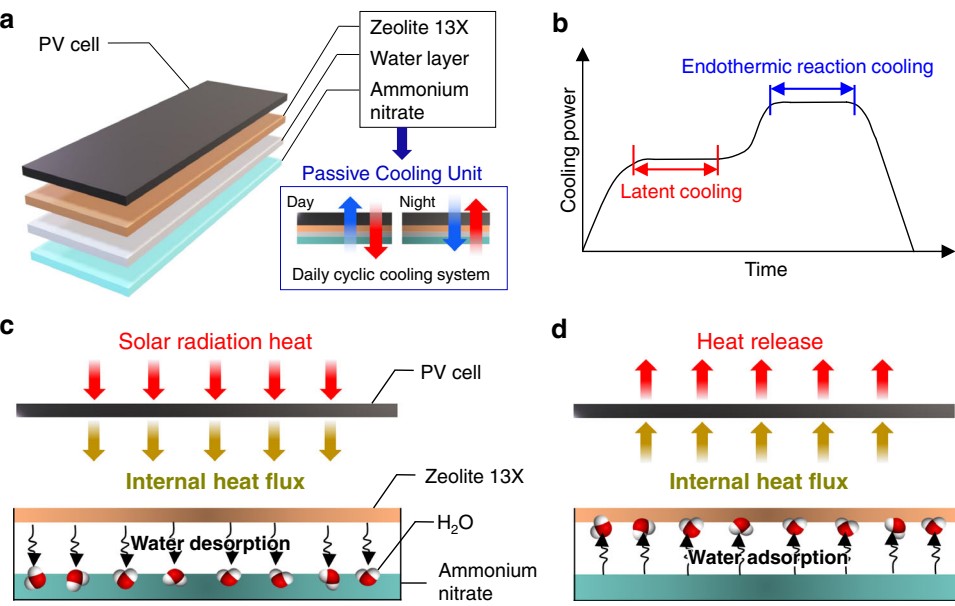

**Fig. 1 | Overview of the passive cooling unit. a** Schematic representation of self-recovering passive cooling unit utilizing the chain reaction of water desorption (primary latent cooling) and dissolution of ammonium nitrate in water (secondary endothermic reaction cooling) integrated with photovoltaic (PV) cell. **b** Cooling power during electricity generation in response to solar radiation. **c** Cooling principle considering internal composition during the day. **d** Self-recovering process during the night.

crystallizes into a solid state (solubility changes owing to the balance in the chemical potential of the solid and aqueous solution states, which is reversible). Finally, liquid $H_2O$ is adsorbed onto zeolite 13X, maintaining $NH_4NO_3$ under dry conditions.

The deeper reflection on the physical working principle of WD-ER unit is described. The water desorption process from zeolite 13X is expressed as Eq. (1), where $Q_d$ is desorption energy (kJ), $m_a$ is adsorbent mass (kg), $h_d$ is average desorption enthalpy of water (kJ/kg-$H_2O$), and $x_w$ is adsorption capacity (kg-$H_2O$/kg-13X). The subscripts $i$, and $f$ represent the initial and final states, respectively.

$$Q_d = m_a \cdot h_d \cdot (x_{w,i} - x_{w,f}) \qquad (1)$$

In the heat exchange process between the PV cell and the thin film (WD-ER unit), the Biot number is less than 0.01, which supports the lumped system analysis in the longitudinal direction, and most of the radiant energy is consumed for water desorption. Moreover, the solar thermal energy is uniformly supplied to the PV modules, enabling one-dimensional analysis. The well-known diffusion equation of desorbed water is Eq. (2)[31], where $m_w$ is desorbed water mass (kg), $K^*$ is modified mass transfer coefficient (m/s), $A$ is cross-sectional area (m²), $C_s$ is concentration of water on the particle surface (kg/m³), and $C_f$ is free stream concentration (kg/m³), respectively (This equation is defined as a pseudo first-order kinetic model[35]).

$$\frac{d(m_w)}{dt} = K^* \cdot A \cdot (C_s - C_f) \qquad (2)$$

However, in the case of WD-ER unit, the diffusion of water molecules is limited in that the concentration difference is not significant. Although the desorption enthalpy of water from microporous materials is lower than that of water vaporization, the cooling effect is estimated as high as 27.4 kJ/mol-$H_2O$.

The endothermic reaction cooling of dissolution process is estimated by Eq. (3), where $Q_e$ is endothermic reaction energy (kJ) and $x_s$ is adsorption capacity (kg-$NH_4NO_3$/kg-$H_2O$). The subscript $s$ represent solute.

$$Q_e = m_w \cdot h_e \cdot (x_{s,f} - x_{s,i}) \qquad (3)$$

The dissolution enthalpy ($h_e$; kJ/kg-solute) is the difference between the formation enthalpies of the aqueous and solid phases. The ideal solubility is defined as the saturated mole fraction of the solute in the solution. It is evaluated by the equilibrium of the chemical potential between the solid and solution states as Eq. (4), where $y$ is the mole fraction of solute in the solution, $R$ is the gas constant, $T$ is temperature, and $T_F$ is the freezing temperature of the solute, respectively.

$$\ln(y) = \frac{h_e}{R}\left(\frac{1}{T} - \frac{1}{T_F}\right) \qquad (4)$$

The total cooling energy density of WD-ER unit can be obtained by considering the chain reaction of water desorption and solute dissolution.

## Porous materials for latent cooling

The economic feasibility of WD-ER materials is required for industrial applications of PV cooling units. Therefore, different types of zeolites are considered as micro- and mesoporous materials applied at the back of the PV cell. MOFs and covalent-organic frameworks (COFs) cannot be utilized because of their complex linker synthesis and high cost[36]. In particular, MOFs are deformed by $H_2O$ molecules, and are not favorable for the repetition of adsorption-desorption cycles[37]. Finally, zeolites with different pore sizes and chemical structural

characteristics, zeolite 13X, zeolite 5A, zeolite 3A, zeolite Y, silico-aluminophosphate-34 (SAPO-34), and aluminosilicate zeolite-13 (SSZ-13) are considered.

The characterization data for the porous materials are shown in Figure S1. The pore-characteristics are analyzed in Figure S1a of the $N_2$ adsorption curve. Materials with large pore volume have a higher $N_2$ adsorption capacity, and a surface area of 570–980 m²/g is obtained by the Brunauer–Emmett–Teller method. In comparison, the $N_2$ adsorption capacity of Zeolite 3A, which has a pore size smaller than 5 Å, is noticeably low. This is related to the low selectivity of gas by considering the size of $N_2$ molecules (1.55 Å). From the XRD pattern presented in Figure S1b, peaks of scattering angle are observed at 2θ = 6.1, 10.0, 11.7, 15.4, 20.0, 23.3, 26.6, 30.9, and 33.6°. Based on that, the porous materials comprising sodium, aluminum, oxygen, silicon, and hydrogen have similar bonds. Although porous materials have similar chemical bonds and components, different pore/crystal structures are created by different synthesis methods that generate different adsorption behaviors.

The micropore (0.5–2.0 nm) is scanned as illustrated in Figure S1c, and zeolites 13X and 5A present a pore size distribution of 0.5–1.0 nm. The pore size of zeolite 3A is estimated to be smaller than 0.5 nm (the total volume of zeolite 3A is high and contains micropores. However, it is not detected in micropore scans of 0.5 nm or more). Zeolites 13X and 3A present large mesopores as illustrated in Figure S1d. FE-SEM images of porous materials are illustrated in Figure S2, and there is no distinction between their macrostructures. A single particle has a hexahedron structure with a size distribution of 0.3–4 µm. The distribution of pores can be estimated from the angular macrostructure (Figure S2). The components of the synthesized porous materials are similar to the theoretical chemical compositions, as determined by energy-dispersive X-ray spectrometry (Table S1).

The water sorption performance can be examined by adsorption-desorption experiments with respect to the relative pressure of water. The water adsorption capacity is analyzed at an isotherm of 20 °C and 40 °C, and the linear correlation of capture capacity is obtained for temperature. The working capacity of water during the latent cooling process is estimated between 20 °C and 70 °C, considering that the temperature of the PV cell rises to approximately 70 °C. The working capacity (kg-medium/kg-carrier) means a mass of reacting medium that can produce cooling effect in the operating temperature range of 20–70 °C per unit mass of carrier, i.e., in latent cooling, the amount of desorbed water (medium) from the unit mass of porous material (carrier), and in endothermic reaction cooling, the amount of dissolved solute (medium) in a unit mass of solvent (carrier). On the other hand, the cooling energy density (kJ/kg-carrier) can be obtained by average reaction/adsorption heat multiplied by working capacity.

The isothermal water adsorption capacity is proportional to the surface area of the porous materials. Note that, although the water adsorption capacity is high at a specific temperature, when the regeneration temperature is high (i.e., the working capacity of water is low because the decrease in adsorption capacity is small as the temperature rises), it is not suitable for cooling applications with a small latent cooling energy density. Specifically, as heat is supplied, the desorption of water molecules must be induced by pore deformation, rather than the kinetic energy increase of water molecules. The water adsorption curves of porous materials at 20 °C and 40 °C are significantly different in Figure S3. Zeolite 13X has the largest working capacity (0.191 g-medium/g-carrier) as illustrated in Fig. 2a. The water adsorption capacity of zeolite Y is the highest at 20 °C, as illustrated in Figure S3c. However, it cannot absorb large amounts of latent heat for the operating temperature range, because there is no significant change in capacity even when the temperature increases. As illustrated in Figure S3d and S3e, the working capacities of SAPO-34 and SSZ-13 are substantially low.

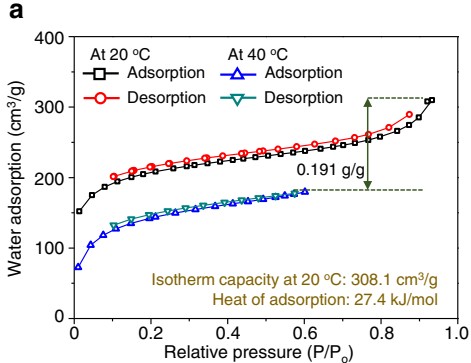
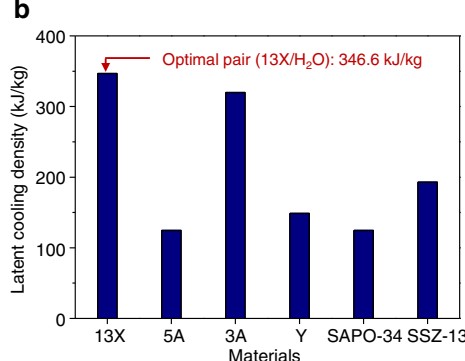

**Fig. 2 | Latent cooling behavior of porous materials. a** Water sorption performance of zeolite 13X. **b** Latent cooling energy density considering water sorption capacity and heat of reaction. Source data are provided as a Source data file.

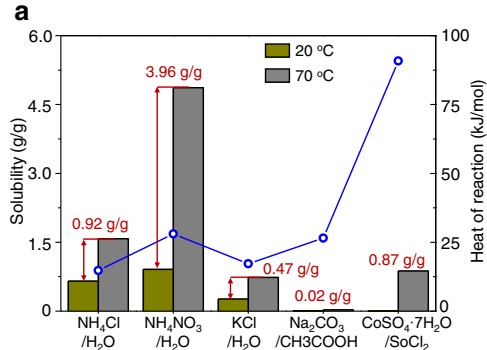
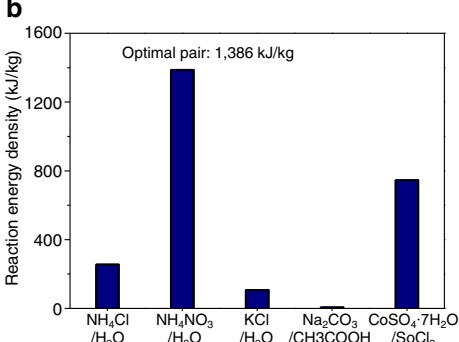

**Fig. 3 | Theoretical analysis of endothermic reaction pairs for water desorption-driven endothermic reaction (WD-ER) cooling. a** Ideal solubility and heat of reaction. **b** Endothermic reaction cooling energy density. Source data are provided as a Source data file.

The heat of reaction (kJ/mol·medium) is estimated during the water sorption process of the isotherm, as illustrated in Figure S4, and it varies in terms of water loading. However, the maximum heat of adsorption is required for the desorption of water molecules, and the latent cooling energy density is obtained from this characteristic. Zeolite Y has the highest adsorption heat of 38.7 kJ/mol-$H_2O$, and other materials provide 25–30 kJ/mol-$H_2O$, which implies physical adsorption between water molecules and porous materials. In addition, the latent cooling energy density is analyzed based on the working capacity and maximum adsorption heat, as illustrated in Fig. 2b. As Zeolite 13X can generate 346.6 kJ/kg-carrier of cooling energy, zeolite 13X/$H_2O$ pair is optimal for latent cooling utilizing the sorption-desorption phenomenon.

## Solvent pairs for endothermic reaction cooling

The best pairs for endothermic reaction cooling are discussed by analyzing the dissolution phenomenon of the solute in the solvent based on Eqs. (3) and (4). The solubility difference at an operating temperature of 20–70 °C is illustrated in Fig. 3a and Table S2. $NH_4NO_3$/$H_2O$ has the highest working capacity of 3.96 g-$NH_4NO_3$/g-$H_2O$ (g-medium/g-carrier) and the reaction heat of $CoSO_4 \cdot 7H_2O$/$SOCl_2$ is the highest at 90.8 kJ/mol-medium; however, the working capacity is minimal, and the endothermic reaction cooling energy density is low, as illustrated in Fig. 3b. The cooling energy density of $NH_4NO_3$/$H_2O$ is 1,386 kJ/kg-carrier.

In summary, the zeolite 13X/$H_2O$ pair is adequate for performing latent cooling, and considering the working capacity, a mass ratio of 1:0.25 is suitable. When the solvent is added more slightly than the working capacity of the zeolite 13X, WD-ER will maintain a humid state at ambient temperature. This decreases the thermal contact resistance as heat is transferred to the liquid. The regenerated liquid

water dissolves the $NH_4NO_3$ layer, and secondary endothermic reaction cooling occurs. The mass ratio of water to ammonium nitrate is 0.25:0.99, considering the ideal solubility.

## Lab-scale heat dissipation performance using WD-ER

The heat dissipation process is visualized to clarify the working mechanism of WD-ER based on the chain reaction in Figure S5. The saturated zeolite 13X absorbs heat to desorb water, as illustrated in Figure S5b. The temperature of the reactor is maintained at approximately 70 °C by the latent cooling effect, while the surface temperature of heating plate corresponds to 200 °C. It is observed that zeolite 13X at the bottom is completely desorbed by the change in color. In the upper part of the completely desorbed layer, liquid water is generated from zeolite 13X in the temperature range of 30–80 °C, which blocks thermal dissipation. Although the temperature of 100 °C or higher is required to completely desorb water from zeolite 13X[38], it has been verified that some water is regenerated at an operating temperature (30–70 °C) based on the working capacity as discussed in Fig. 2. In Figure S5c, the endothermic reaction cooling occurs with the dissolution of $NH_4NO_3$ in water as the temperature rises with the heat supply.

The heat dissipation performance is evaluated with Figure S6. The simple energy balance is defined in Eq. (5) by analyzing the cell as the control volume to compare the cooling methods. $m$ is the mass of the cell (kg), $C_p$ is the specific heat capacity of the cell (kJ/kg·K), $\dot{Q}$ is the supplied heat transfer rate (kW), $\Delta t$ is the time interval between steps (s), $h_{eff}$ is the effective heat transfer coefficient (W/m²·K), $A$ is the cross-sectional area (m²), $T_{s,i}$ is the cell temperature at the $i$th interval (°C), and $T_a$ is the ambient temperature (°C).

$$m \cdot C_p \cdot (T_{s,i+1} - T_{s,i}) = \dot{Q} \cdot \Delta t - h_{eff} \cdot A \cdot (T_{s,i} - T_a) \cdot \Delta t \quad (5)$$

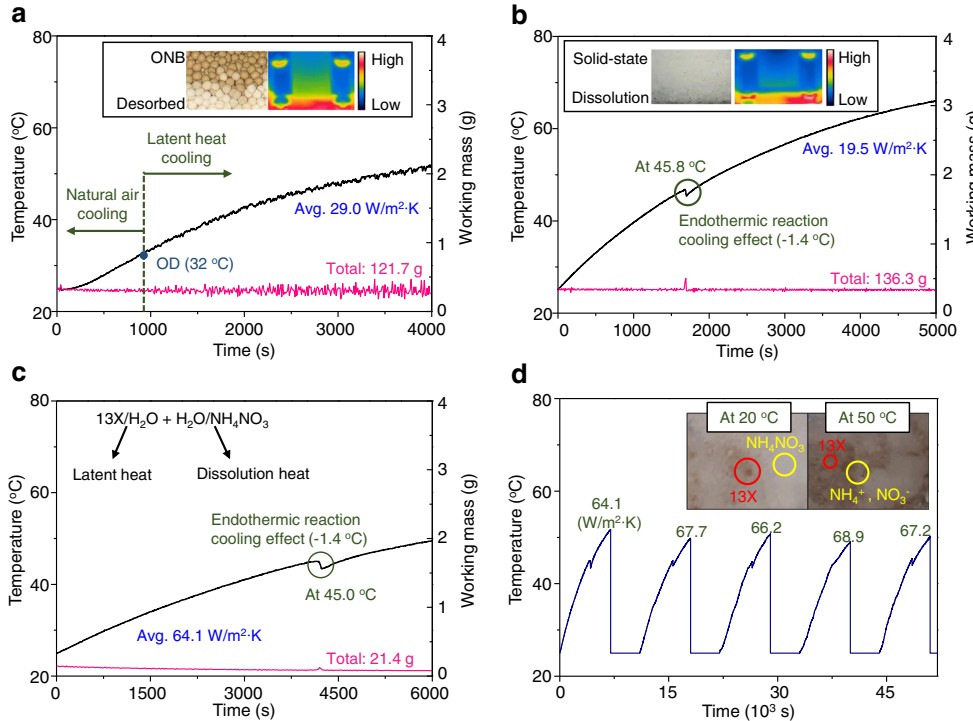

**Fig. 4 | Heat dissipation performance in terms of cooling methods. a** Latent cooling. **b** Endothermic reaction cooling. **c** Water desorption-driven endothermic reaction (WD-ER) cooling. **d** Repeated WD-ER cooling. OD onset of desorption. Source data are provided as a Source data file.

Although the WD-ER cooling unit extracts thermal energy by latent heat of porous material and endothermic reaction of solute layer, a convective heat transfer model utilizing effective heat transfer coefficient of Eq. (5) is applied for quantitative comparison. Meanwhile, the radiant heat transfer can be neglected in that the heat dissipation experiment was conducted indoors. However, the operation of WD-ER cooling unit is invariant regardless of the form of external heat transfer. The Biot number of natural air cooling is $4 \times 10^{-5}$ which makes it possible to neglect the spatial temperature distribution.

The temperature variation of the PV cell with natural air cooling is illustrated in Figure S7a. The air around the cell is heated, which leads to natural convection induced by the density difference of air. The effective heat transfer coefficient of natural air cooling is 1.04 W/m²·K, which is very low, and the temperature consistently rises because of the constantly supplied heat (in general, the heat transfer coefficient of natural air cooling is less than 10 W/m²·K). The natural convection of air can be characterized by the Rayleigh number in Eq. (6). $\Delta\rho$ is the density difference owing to temperature variation (kg/m³), $l$ is the characteristic length of the cell (m), $g$ is the gravitational acceleration (m/s²), $\mu$ is the dynamic viscosity of air (Pa·s), and $\alpha$ is the thermal diffusivity (the ratio of thermal conduction to absorbed heat; m²/s).

$$Ra = \frac{\Delta\rho \cdot l^3 \cdot g}{\mu \cdot \alpha} \quad (6)$$

The order of $Ra$ is $10^5$ in the present study, and the Nusselt number correlation is given by Eq. (7). The theoretical heat transfer coefficient is similar to the experimental results in Figure S7a, which validates the estimation of the cooling performance

$$Nu_n = 0.54 \cdot Ra^{\frac{1}{4}} \quad (7)$$

Figure S7b illustrates the temperature variation of forced air convection where the flow of air is generated by a fan (average air velocity is 1.96 m/s). The temperature gradient is lower than that of

natural air cooling, and is saturated at approximately 61 °C, which indicates that heat dissipation by forced air cooling and supplied heat is at thermal equilibrium. The Reynolds number of forced air cooling is $1.2 \times 10^3$, the Prandtl number is 0.7, and $Nu$ can be calculated using Eq. (8). The theoretical heat transfer coefficient is 12.1 W/m²·K, which has a 22% error, compared to the experimental result.

$$Nu = 0.664 \cdot Re^{0.5} \cdot Pr^{\frac{1}{3}} \quad (8)$$

t is possible to enhance the heat transfer coefficient 10 times by forced convection, compared to natural air cooling, and the temperature gradient significantly decreases. However, the cooling effect deteriorates when the ambient temperature increases (note that this experiment was conducted at an ambient temperature of 20 °C). Moreover, cooling with a fan is difficult for application in building-integrated PV cells because of energy consumption, noise generation, and requirement of additional spatial area, which reduces power generation per area. The difference between natural and forced air cooling is reduced in industrial applications, considering the turbulent flow due to the intermittent wind from the outside.

In Fig. 4a, the heat dissipation performance of the latent cooling is evaluated by the thin film of water-saturated zeolite 13X coated to PV cell. The temperature gradient is significantly lower than that of forced air cooling. When the temperature is lower than 32 °C, heat is absorbed to increase the temperature of the porous structure. The temperature fluctuation occurs after 32 °C, which indicates that the supplied heat is absorbed by the porous structure, and is transferred to water desorption unevenly. Specifically, the onset of desorption (OD) occurs at 32 °C, i.e., a temperature difference of 12 °C is the minimum potential to induce the desorption. The effective heat transfer coefficient of latent heat cooling is estimated as 29.0 W/m²·K, which is higher than that of forced air cooling. Note that, forced convection is an active cooling method in which a consistent cooling effect can be generated. However, cooling with porous materials is possible until all the solvent is regenerated based on the working capacity. The mass of water

desorbed from zeolite 13X, which is defined as working mass ($\Delta m_w$), is calculated using Eq. (9), where $h_d$ is the water desorption enthalpy obtained from Figure S4. $h_{N,A}$ is heat transfer coefficient of natural air cooling.

$$m \cdot c_p \cdot (T_{s,i+1} - T_{s,i}) = \dot{Q} \cdot \Delta t - h_d \cdot \Delta m_w - h_{N,A} \cdot A \cdot (T_{s,i} - T_a) \cdot \Delta t$$
$$(9)$$

In the present study, water (122 g) is desorbed for 4000 s of the cooling process. WD-ER can be optimally designed by considering the cooling load of the PV cell, based on the working capacity of water.

The heat dissipation of the endothermic reaction cooling is evaluated in Fig. 4b, utilizing a pair of $NH_4NO_3/H_2O$ at an ideal mass ratio, which was attached to PV cell as a coating layer. The effective heat transfer coefficient is 19.5 W/m²·K, which is slightly lower than that of latent cooling. The working mass of $NH_4NO_3$ owing to increase in temperature is 136.3 g for 4000 s. The calculation method for the working mass is equivalent to that in Eq. (9) by considering the endothermic reaction enthalpy instead of desorption enthalpy. A sudden temperature drop occurs at 45.8 °C. Below this point, a small amount of $NH_4NO_3$ crystals reacted with water to generate endothermic heat. However, the $NH_4NO_3$ layer collapses at that temperature, leading to a drastic cooling effect, which is related to the solubility of $NH_4NO_3$.

As presented in Fig. 1, the cooling performance of WD-ER can be maximized by a combined structure comprising a latent cooling layer of porous materials and an endothermic reaction cooling layer of ammonium nitrate crystals that is driven by water as a solvent. The heat dissipation performance of the WD-ER cooling unit is illustrated in Fig. 4c. The water-saturated zeolite 13X was coated on a PV cell as a thin film, and a dispersed $NH_4NO_3$ crystal layer was attached, which presented the same structure with practical applications. The temperature gradient of WD-ER cooling unit is the lowest compared to other cooling methods. A sudden temperature drop at 45 °C is also observed, as in the case of the $NH_4NO_3/H_2O$ pair. The mass of the solvent required for operation can be estimated using Eq. (9), considering the ideal solubility and dissolution enthalpy of $NH_4NO_3$ and the desorption energy of water. Because the cooling of WD-ER is a two-step process of water desorption from zeolite 13X and dissolution of $NH_4NO_3$ in water, the working mass of the solvent is significantly less than the mass required for the latent cooling or endothermic reaction cooling method. More importantly, the average effective heat transfer coefficient of WD-ER is 64.1 W/m²·K. In other words, the cooling heat flux reaches 1923 W/m², when the temperature difference is considered as 30 °C between the device and ambient, which can be applied to high power electronics such as semiconductor devices[39].

To further discuss the heat dissipation mechanism, the average effective heat transfer coefficient of the WD-ER unit ($13X/H_2O/NH_4NO_3$) is 64.1 W/m²·K, far superior to the sum of effective heat transfer coefficients of latent heat cooling utilizing $13X/H_2O$ (29 W/m²·K) and endothermic cooling utilizing $NH_4NO_3/H_2O$ (19.5 W/m²·K). The latent cooling mechanism of WD-ER is invariant compared to that of $13X/H_2O$ in that the thermal energy supplied to the PV is transferred to the thin film and it induces water desorption. However, the WD-ER presents a different aspect in terms of dissolution process when only endothermic reaction cooling ($NH_4NO_3/H_2O$) is performed. As thermal energy is directly supplied to $NH_4NO_3/H_2O$, the solubility increases and $NH_4NO_3$ is dissolved, resulting in an endothermic reaction. In this process, the reaction area between $NH_4NO_3$ and $H_2O$ is limited in that water cannot flow between the $NH_4NO_3$ crystal layers. When endothermic cooling is induced after the latent cooling for WD-ER, water molecules generated in the form of droplets permeate between the $NH_4NO_3$ crystal layers and the reaction area widens (in Figure S5b, the water droplets are generated on the surface of zeolite 13X). In

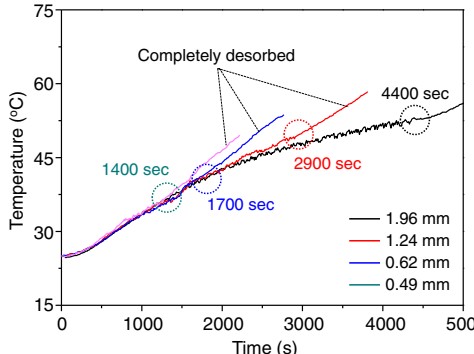

**Fig. 5 | Heat dissipation performance of latent cooling in terms of zeolite film thickness.** Source data are provided as a Source data file.

summary, the cooling power of WD-ER increases due to the improvement of dissolution process by widening the interfacial area. On the other hand, the cooling energy density of WD-ER (2876 kJ/kg-$H_2O$) is similar to the sum of cooling energy densities of latent cooling (346.6 kJ/kg-13X is equivalent to 1815 kJ/kg-$H_2O$) and endothermic reaction cooling (1386 kJ/kg-$H_2O$) with a 10% error.

It is evaluated that after the WD-ER cooling unit performs heat dissipation, whether all processes are self-recovered by cyclic experiments. In Fig. 4d, as the effective heat transfer coefficient is maintained at 64.1–68.9 W/m²·K in five cycles, it is concluded that the WD-ER cooling unit exhibits self-recovering characteristics. Specifically, at 20 °C, zeolite 13X completely adsorbs water again, $NH_4NO_3$ crystallizes to form a hard layer, and WD-ER exhibits a solid-like structure. When heat is supplied, water is regenerated from zeolite 13X, and the water dissolves $NH_4NO_3$ again to generate a cooling effect. Accordingly, at 50 °C, an aqueous solution in which $NH_4^+$ and $NO_3^-$ are dissolved is observed, and the color of zeolite 13X is pale, indicating that water is desorbed. Note that, in the solidification process of $NH_4NO_3$, crystals could be randomly formed. However, the reproducibility problem due to the random crystallization does not occur in Fig. 4d because the interfacial area of dissolution is large as the water droplets desorbed from zeolite 13X permeate between them. Moreover, the thickness change of zeolite 13X film layer does not affect the heat dissipation kinetics (rate/behavior), and only the cooling energy density is varied as presented in Fig. 5. As the temperature change is the driving force of water desorption, a certain amount of water molecules are regenerated regardless of the thickness of the layer (The rate-limiting step of water desorption process is dominated by the cross-sectional area instead of the thickness)[31]. As the thickness of zeolite 13X film is in the range of 1–5 mm, the thermal resistance can be neglected with the Biot number of $1.5$–$7.5 \times 10^{-5}$.

**Practical applications of WD-ER for photovoltaic cell cooling**
The WD-ER cooling unit is applied to a PV cell comprising mono-crystalline silicon. As illustrated in Fig. 6a, urethane waterproofing on the back of the PV cell protects the power terminal unit from moisture penetration. Zeolite 13X particles are coated on the PV cell with a thickness of 2 mm, and a negligible amount of polyvinyl acetate is included. After the coated zeolite 13X is saturated with water, the $NH_4NO_3$ crystal layer is added to configure the WD-ER cooling unit (The configuration of PV cell with WD-ER cooling unit for industrial applications is same as that of lab-scale heat dissipation experiment in Fig. 4). PV cells with natural air cooling and WD-ER cooling units were installed in adjacent positions to observe the temperature variation. Based on the images taken with a thermal imaging camera at 2 pm, the surface temperature of the PV cell with the WD-ER cooling unit is 39–43 °C, which is significantly lower than that of natural air cooling (51–56 °C) as shown in Fig. 6a. Note that

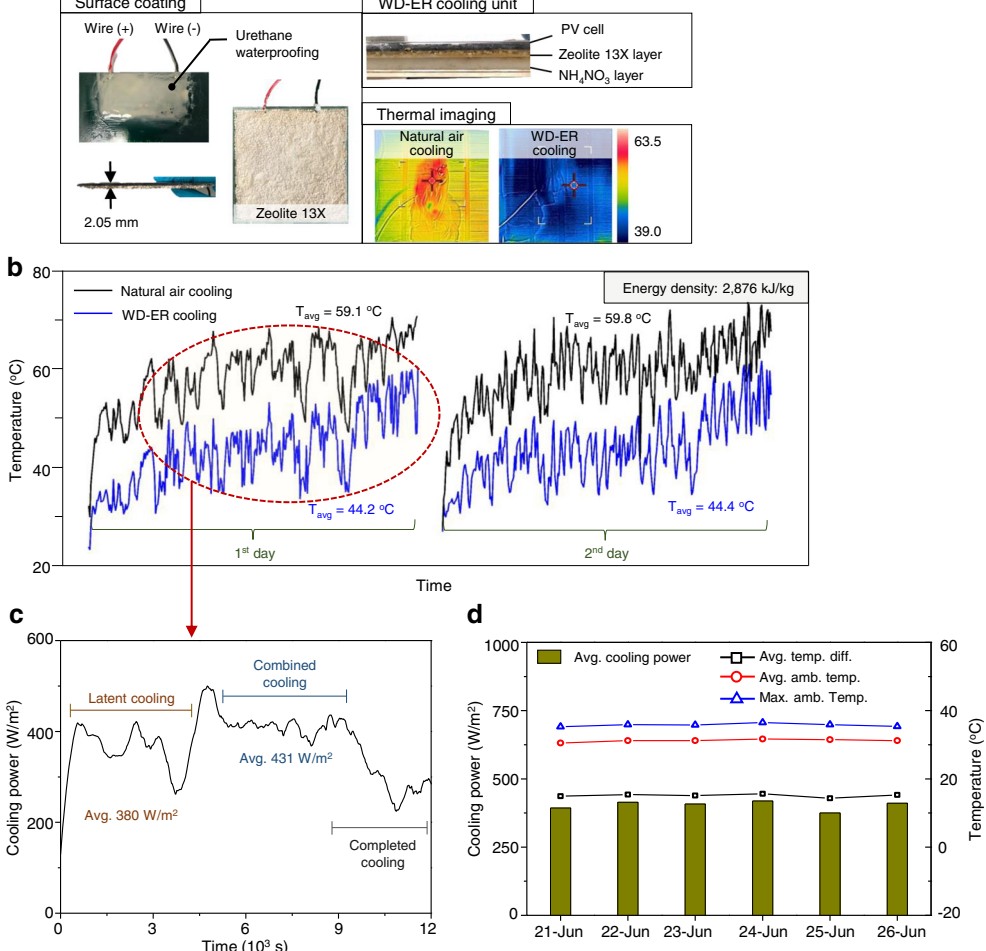

**Fig. 6 | Practical application of water desorption-driven endothermic reaction (WD-ER) cooling unit for photovoltaic (PV) cell. a** Images of PV cell with WD-ER unit. **b** Temperature variation of PV cell during a day. **c** Net cooling power density of WD-ER unit without natural convection. **d** Practical cyclic working performance of WD-ER unit. Source data are provided as a Source data file.

although the tape for attaching the thermocouple has a higher thermal absorptivity than monocrystalline silicon, the effect on the temperature field of PV cell can be neglected in that it accounts for only 8% of the total area. When the temperature of PV cell with the tape is analyzed by Fourier's law, the temperature deviation from other PV cell parts is within 0.3 °C.

Figure 6b presents the temperature variation during the power generation process of the PV cell. The severe temperature fluctuation results from the phenomena such as intermittent winds inducing local forced air cooling and obscuring of sunlight by clouds. The temperature of the PV cell with WD-ER is much lower than that of natural air cooling, and the average temperature is reduced by 14.9 °C (from 59.1 °C to 44.2 °C). Figure 6c illustrates the net cooling power density considering the temperature difference between natural air and WD-ER cooling (temperature data were converted into cooling energy per unit area of PV cell utilizing the smoothing filter of Savitzky-Golay theory. After dividing it by the measurement time interval, the cooling power density was obtained). When the cell temperature is in the range of 20–45 °C (time range of 0–3500 s), the radiation heat is absorbed at 380 W/m², which corresponds to the desorption energy of water from zeolite 13X (latent cooling). The absorption heat amount rapidly increases to 495 W/m² at 45 °C as the $NH_4NO_3$ layer reacts with solvent, which is the endothermic reaction cooling for 3500–4000 s. It is similar to the case when WD-ER cooling unit is applied for lab-scale

heat dissipation experiment in Fig. 4c, the temperature rapidly decreases around 45 °C. Further, the cooling power density is maintained at 431 W/m² with the combined water desorption and solute dissolution. When the cooling energy density reaches 2500 kJ/kg at approximately 9000 s, the cooling performance is significantly reduced as the WD-ER cooling unit begins to be saturated. Finally, the cooling energy density is 2876 kJ/kg, which is sufficient for utilization under high ambient temperature conditions.

The cyclic cooling performance of WD-ER is evaluated in Fig. 6d, when the outdoor temperature is 30.5–31.7 °C. During the power generation period, the temperature of PV cell with WD-ER reaches a maximum of 60 °C, and after the sunset, the temperature drops to around 25 °C, confirming the self-recovering characteristics (a temperature difference of 35 °C, which is maintained for more than 8 h, is sufficient for recovering process. In addition, the radiation cooling effect is also significant at night). The average cooling power density is uniform at 375–419 W/m², although it slightly differs in terms of external conditions. In winter, the cooling effect of the WD-ER unit could be more important as the working temperature range becomes larger (In general, the temperature range of PV cell is 0–55 °C during power generation, and at night, the temperature drops to −10 °C). Figure S8 illustrates the current-voltage characteristics of the monocrystalline silicon. As the temperature rises, the open-circuit voltage increases, while the short-circuit current decreases. The

**Table 1 | Comparison of WD-ER cooling unit with passive cooling methods**

| Methods | | Density (kJ/kg) | Conditions and remarks | Ref |
|---|---|---|---|---|
| WD-ER | 13X/H$_2$O/NH$_4$NO$_3$ | 2876 | Operating range: 20–70 °C<br>Cooling power: 403 W/m$^2$<br>Temperature drop: 15.1 °C | |
| Phase change materials | RT55 | 170 | Melting: 51–57 °C | 40 |
| | RT28HC | 220 | Melting: 28 °C | 41 |
| | C-PCM | 199 | Melting: 21.8 °C | 42 |
| | OM47 | 196 | Melting: 48 °C | 43 |
| | MF-PW30 | 139.8 | Melting: 56.8 °C, crystallization: 45.1 °C | 44 |
| | PCMB | 141.7 | Steady range: 40.4–84.9 °C | 45 |
| | GO-HS/PAAAM | 200.3 | Melting: 23 °C | 46 |
| | Eicosane | 237.4 | Melting: 36.5 °C | 47 |
| | Tricosane | 269.2 | Melting: 42–48 °C | 48 |
| | MIL-101(Cr) | 1950 | Operating range: 20–70 °C | 31 |
| | PAN-CNT-CaCl$_2$ | – | Operating range: 20–70 °C<br>Cooling power: 295 W/m$^2$<br>Temperature drop: 10.0 °C | 18 |
| Radiative cooling | Cooling assembly system | – | Operating range: 74 °C<br>Cooling power: 310 W/m$^2$<br>Temperature drop: 36.6 °C | 49 |
| | Silica pyramid | – | Operating range: 20–70 °C<br>Temperature drop: 18.3 °C | 50 |
| | Multilayer photonic film (Al$_2$O$_3$/SiN/TiO$_2$/SiO$_2$) | – | Operating range: 40–70 °C<br>Cooling power: 149 W/m$^2$<br>Temperature drop: 5.7 °C | 19 |
| | Multilayer stack (SiO$_2$/TiO$_2$/MgF$_2$) | – | Operating range: 10–30 °C<br>Cooling power: 29.5 W/m$^2$ | 51 |

maximum power is inversely proportional to temperature (25 °C: 91.3 mW → 60 °C: 81.4 mW). In summary, the power efficiency of PV cell with WD-ER unit is improved by approximately 10% compared to that of PV cell with natural air cooling. The improvement of power efficiency has been analyzed based on a commercially available monocrystalline silicon.

Table 1 compares the cooling performance of WD-ER unit with other methods reported in the literature[18,19,31,40–51]. The cooling energy density of PCMs ranges 100–300 kJ/kg. Although the temperature of PCMs is maintained at the melting point during the cooling process, the sensible heat change of the PV cell occurs continuously (i.e., the temperature of PV cell is not kept constant regardless of the state of PCMs). Moreover, the cooling energy density of MIL-101(Cr) coating layer is high at 1950 kJ/kg using ambient moisture sorption-desorption process for electronic device applications[31]. Similarly, the PV cooling by atmospheric water sorption-evaporation cycle is proposed, and the cooling power is 295 W/m$^2$ with temperature drop of 10 °C. The radiative methods present cooling power of 40–300 W/m$^2$. The WD-ER of the present study outperforms the literature cooling methods in terms of cooling energy density (2876 kJ/kg) and cooling power (403 W/m$^2$).

## Discussion

Inspired by recent studies on passive cooling strategy for PV cell, which improves power generation efficiency and durability of the cell, a self-recovering passive cooling unit comprising inexpensive materials without maintenance problems is developed. It is defined as a water desorption-driven endothermic reaction (WD-ER) cooling unit. The WD-ER cooling unit comprises a water-saturated zeolite 13X and an ammonium nitrate crystal layer to form a thin film. The water sorption performance of porous materials is inversely proportional to temperature, and the solubility of ammonium nitrate in the solvent increases with temperature. When thermal energy is supplied, the temperature of zeolite 13X increases, heat is absorbed for water

desorption in the pores (primarily latent cooling; 346.6 kJ/kg-carrier), and liquid water dissolves ammonium nitrate crystals (secondary endothermic reaction cooling; 1386 kJ/kg-carrier). By the chain reaction of water desorption and solute dissolution, the overall cooling energy density of the WD-ER unit reaches 2876 kJ/kg, and it can be optimized by considering the required cooling energy density. When the temperature is lowered at night, it is self-recovered, such that the ammonium nitrate is crystallized and water is adsorbed into zeolite 13X.

For industrial applications of WD-ER cooling unit with PV cell composed of monocrystalline silicon, the temperature of the cell can be reduced by 15.1 °C during the day. In the temperature range of 20–45 °C, the latent cooling effect is dominant, and the supplied heat is utilized for the desorption of water from zeolite 13X (cooling power: 380 W/m$^2$). In addition, the combined cooling mode of water desorption and dissolution of the solute is induced as the ammonium nitrate layer collapses above 45 °C with the cooling power of 431 W/m$^2$. The average cooling power density is uniform at 375–419 W/m$^2$ during cyclic cooling experiments when outdoor temperature ranges 30.5–31.7 °C. The WD-ER unit outperforms the cooling methods reported in the literature. Note that, as only different types of zeolites are analyzed for WD-ER cooling unit in this study, further improvement would be expected by utilizing advanced approaches including metal-organic frameworks.

## Methods
### Materials preparation
The zeolite X series comprises alumina tetrahedra and silicon oxygen tetrahedra. Space is created by an oxygen bridge, which makes them pore and has a hexagonal prism structure. Zeolite X has the largest pore size and is widely utilized for gas separation (air purification). Zeolite A series comprises sodium and aluminosilicate, similar to zeolite 13X, and has a three-dimensional crystal structure. According to the synthesis method, the pore size is classified as 3A–5A. Zeolite Y

includes silicon or aluminum atoms with $O_4$, creating tetrahedral and sodalite linkers, which comprise the hexagonal prism. SAPO-34 is a chabazite (tectosilicate) structure with 3A and 5A pore sizes. SSZ-13 is also a chabazite topology with a small pore size that is created by an 8 membered ring structure. The synthesis of porous materials can be performed to obtain the desired meso/microstructure by considering the element ratio in Table S1. All reagents were purchased from Sigma-Aldrich. Synthesis of zeolites X, A, and Y: Sodium hydroxide was added to deionized water and dissolved by stirring at 20 °C for 1 h. Further, sodium silicate solution (7.5 wt% $Na_2O$ and 28.5 wt% $SiO_2$) and sodium aluminate (50 wt% $Al_2O_3$ and 37 wt% $Na_2O$) were added and stirred at 40 °C for 24 h. It was heated in a convection oven at 100 °C to form a solid state, and centrifuged with methanol to obtain zeolite 13X in the form of pure powder. Synthesis of SAPO-34: Aluminum oxide hydrated (>64 wt% $Al_2O_3$ basis), phosphoric acid, and sodium silicate solution were added to diethanolamine and stirred at 20 °C for 1 h. A solid state was obtained by maintaining the temperature at 150 °C for 24 h utilizing a convection oven. The volatile substances were removed by heating at 500 °C in a furnace for 3 h and centrifuged with a solvent to obtain pure powder. Synthesis of SSZ-13: Sodium hydroxide and N,N,N-1-trimethyl ammonium hydroxide were added to deionized water and stirred at 20 °C for 1 h. After adding sodium silicate solution (28.5 wt% $SiO_2$) and sodium aluminate (50 wt% $Al_2O_3$), the mixture was maintained at 150 °C for 48 h. Finally, the powder was obtained by washing.

## Characterization

The $N_2$ adsorption curve of the porous materials was evaluated utilizing the Micromeritics 3Flex equipment. Before the gas adsorption experiment, the samples were pretreated in a vacuum at 150 °C for 24 h. The bath temperature was −195.9 °C, and the equilibrium interval was 10 s. Micropores were scanned at 0.5–2.0 nm, and data were collected utilizing the Horvath-Kawazoe differential pore volume method. Pores were investigated at 1.8–300 nm, and calculated by the Barrett–Joyner–Halenda plot method. X-ray diffraction (XRD) was measured with a Rigaku MiniFlex 600 equipment with an X-ray power of 600 W, an X-ray tube with copper, a step size of scattering angle of 0.02°, a scan angle of 5–90°, and a holder diameter of 25 mm. Field emission scanning electron microscopy (FE-SEM) was performed utilizing a Quanta FEG 250 model manufactured by FEI, and pretreated with a Pt target at 20 mA for 120 s. Energy-dispersive X-ray analysis was performed utilizing EDAX's software. Water sorption performance was evaluated by the volumetric method utilizing 3Flex equipment, and it was pretreated by maintaining it under vacuum at 150 °C for 3 h.

## Theory of endothermic reaction cooling

Ammonium chloride/barium hydroxide octahydrate, ammonium chloride/water, cobalt(II) sulfate heptahydrate/thionyl chloride, potassium chloride/water, and sodium carbonate/ethanoic acid, which generate endothermic heat with a reversible process, are considered. The cooling energy density for endothermic reaction is estimated considering the theoretical dissolution energy and ideal solubility of the solute. The heat of dissolution ($q$) can be calculated by the difference between the formation enthalpies of the aqueous and solid phases ($\Delta H$), as expressed as Eq. (10).

$$q = \Delta H \tag{10}$$

The chemical potential of the solid state is expressed as Eq. (11), where $\mu$, $S$, and $T$ represent the chemical potential, entropy, and absolute temperature, respectively.

$$d\mu_{solid} = -S_{solid} \cdot dT \tag{11}$$

The chemical potential of the aqueous state is expressed as Eq. (12), where $R$ is the gas constant, and $y$ is the mole fraction of solute in the solution.

$$d\mu_{solution} = -S_{solution} \cdot dT + R \cdot T \cdot d(lny) \tag{12}$$

Solubility is defined as the saturated mole fraction of the solute in the solution. It is evaluated by the equilibrium of the chemical potential between the solid and solution states of Eq. (13).

$$d(lny) = \frac{S_{solution} - S_{solid}}{RT} dT \tag{13}$$

The entropy change ($S_{solution} - S_{solid}$) of the dissolution process is given by Eq. (14), where $q$ is the reaction heat in Eq. (10).

$$S_{solution} - S_{solid} = \frac{q}{T} \tag{14}$$

The ideal solubility is calculated using Eq. (15), where $T_f$ is the freezing temperature of the solute.

$$ln(y) = \frac{\Delta_{fus}H}{R}\left(\frac{1}{T} - \frac{1}{T_F}\right) \tag{15}$$

## Lab-scale heat dissipation experiments

The heat dissipation process of the passive cooling unit was visualized with the device illustrated in Figure S5a. A constant heat of 700 W was supplied to the reactor, which was maintained at 200 °C for 20 min. Subsequently, the behavior of the materials inside the reactor was observed. E5 from FLIR was utilized for thermal imaging (temperature range of −20 to 250 °C, thermal sensitivity of <0.10 °C, and frame rate of 9 Hz). The cross section of the reactor was a square of 7 cm, and the height of the reactor was 15 mm. The total mass of the materials was 100 g. Figure S6 is an experimental device for evaluating the heat dissipation performance, and a constant heat of 50 W was supplied to the cell at the bottom. A semiconductor cell has a square size of 100 mm and thickness of 5 mm. Thermal grease was applied between the heating plate and the semiconductor cell to reduce the thermal resistance, and the surface temperature of the cell was measured utilizing a thermocouple. Regarding forced air convection, the rotational speed of the fan was 1500 rpm, and when converted into airflow, it was 1.96 m/s. For the latent and endothermic reaction cooling experiments, the materials were attached to the surface of monocrystalline silicon cell in the form of thin film, which is same with industrial applications, and the other conditions were similar for natural air cooling.

## Practical applications

A PV cell (a square size of 14.5 cm) with WD-ER cooling unit is illustrated in Figure S9. The urethane waterproofing coating was applied to protect the power terminal unit on the back of the PV cell, and the thickness was approximately 0.2 mm. Further, 20 wt% zeolite 13X and 1 wt% polyvinyl alcohol (PVA) were dispersed in ethanol. After 8 h at 50 °C in a convection oven, zeolite 13X was coated on the surface of the PV cell, and it was repeated until the 13X coating layer had a thickness of 2 mm. Zeolite 13X was fully saturated by exposure to a moist environment. Subsequently, ammonium nitrate was coated by the same method, and the total mass was adjusted to an optimal composition. Because moisture was generated when heat was absorbed, it was sealed with an acrylic case. The PV cell with the WD-ER cooling unit was placed in a suitable position for power generation, and the cooling performance was estimated during a day. The experiment was conducted from 10 am to 5 pm with an average ambient temperature of 30.5 °C and a maximum temperature of 35.4 °C. The temperature

variation was collected at 10 s intervals utilizing a midi LOGER GL240 manufactured by GRAPHTEC. The current-voltage characteristics according to the temperature of the PV cell were measured with a PROVA-200A manufactured by TES. The experiments were performed while the PV cell was connected to a battery to process the generated electricity.

## Reporting summary
Further information on research design is available in the Nature Portfolio Reporting Summary linked to this article.

## Data availability
All data supporting the findings of this study are available within the article and Supplementary information files. All data are available from the corresponding author upon request. Source data are provided with this paper.

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

## Acknowledgements

This study was supported by a National Research Foundation of Korea (NRF) grant funded by the Korean government (MSIT) (No. NRF-2020R1A5A1018153 and No. 2022R1A2B5B03002421).

## Author contributions

S.K. performed data curation, investigation, experimental/theoretical analysis, and writing. J.H.P. conducted data curation and experimental analysis. J.W.L. conducted data curation, discussion, and theoretical analysis. Y.K. carried out discussion, outlining, and writing-review. Y.T.K. supervised conceptualization, methodology, writing review, and editing.

## Competing interests

The authors declare no competing interests.
