## [Peer Review File · Nature Communications]

Self-recovering passive cooling utilizing endothermic reaction of $\text{NH}_4\text{NO}_3/\text{H}_2\text{O}$ driven by water sorption for photovoltaic cellReviewers' comments:

Reviewer #1 (Remarks to the Author):

This manuscript reports a solar cell cooling strategy based on the endothermic reaction of $\text{NH}_4\text{NO}_3/\text{H}_2\text{O}$ driven by moisture sorption. Below are my comments:

1. First of all, a very important and related reference [Photovoltaic panel cooling by atmospheric water sorption–evaporation cycle, *Nature Sustainability* 3, 636–643 (2020)] is missing. This work should be in the discussion of prior work. The novelty of the proposed approach, as compared to the existing work, should be clearly presented,
2. In addition, in the introduction discussion of the existing approach, references of radiative cooling are also missing, for example [A Comprehensive Photonic Approach for Solar Cell Cooling, *ACS Photonics* 4, 774–782 (2017)]. In addition, the use of solar cell cooling at night for water sorption has also been proposed [Nighttime Radiative Cooling for Water Harvesting from Solar Panels, *ACS Photonics* 8, 269–275 (2021)], which is also related to the proposed approach.
3. The authors claim “Herein, a novel type of passive cooling unit for a PV cell is proposed, and presents the best performance among cooling methods that have been reported in the literature^{19,20}”. However, in the experiments part, the main comparison is with natural air cooling. A comparison of the proposed approach with all existing approaches should be presented, to support their claim. Also, references 19, 20 do not serve as supporting references in this context.
4. The writing and organization of this manuscript should also be significantly improved. The first two paragraphs are too generic. They should be more related to the cooling of solar cells. Too many figures in the main text. Many figure data in the main text are just supporting materials characterization data. They did not serve as direct proof of the main claim. They should be reorganized or moved to supporting information.
5. While too many supporting data are presented in the manuscript, the key data is insufficient. For example, the cooling effect of this approach (Figure 8b) should have at least shown a few full-day cycles to show it can self-sustainably work with the daytime and nighttime working mechanisms.

Reviewer #2 (Remarks to the Author):

The manuscript elaborates the use of moisture sorption driven endothermic reaction (MS-ER)-based cooling of the photovoltaic cells for reduction in photovoltaic power generation costs over a 10-year period and increase in energy production density.

The MS-ER cooling is achieved using a water saturated zeolite film and an ammonium nitrate film under the photovoltaic cell. The increase in PV cell temperature leads to water desorption from the zeolite which results in latent cooling. The expunged water vapours dissolve ammonium nitrate crystals which results in further endothermic cooling. When the PV cell cools at night it was claimed that the reduction in temperature leads to precipitation of ammonium nitrate crystals, and water is adsorbed by the zeolite layer. This is the claimed self-healing effect.

Overall, the manuscript presents good scientific discussion, good figures, logical development of arguments, good combination of experimental and numerical approaches and commendable novelty in the use of porous materials as water reservoirs. However, the research still lacks critical elements essential for qualifying into an esteemed science journal such as Nature Communications.

These issues are described below:

1) The concept of photovoltaic cell cooling for improvement in cell efficiency is now well proven. Different approaches have been demonstrated for reduction of PV cell temperature such as air cooling, use of cooling fins, use of microchannels, recirculating fluid cooling (water/oil/others), cooling induced by water spray, cooling induced by Peltier cells and cooling induced by the use of phase change materials. For instance, reviews of such methods are available in the literature. (<https://doi.org/10.1016/j.matpr.2020.07.130>, <https://doi.org/10.1016/j.solener.2021.01.016>)

2) The manuscript compares the performance of the presented MS-ER technique against air-cooled PV cells which is not a fair comparison. It is known that air-cooling of photovoltaic cells is the least efficient method. A good comparison would have been against phase change materials such as RT55, RT28HC, OM29, OM-47 etc. (<https://doi.org/10.1016/j.matpr.2020.07.130>) MS-ER is basically a combination of moisture sorption/desorption (similar to phase change) and dissolution. Thus, comparison should be done against phase change materials.

3) Compared to air-cooled photovoltaic cells only marginal improvements were achieved such as reduction in power generation cost by 5% over a period of 10 years and improvement in power efficiency by 7.5%. Thus, the manuscript needs to answer the big question "Why should MS-ER approach be employed in the first place when phase change materials result in better power efficiencies and

benefits?" The easiest way to answer this critical question is to add a table of comparison in the manuscript which compares the MS-ER method against other PV cell cooling methods such as the use of phase change materials, use of cooling fluids and use of Peltier cells.

4) The manuscript claims self-healing passive cooling using endothermic reaction. However, the self-healing aspect was not demonstrated. It is ideal that the authors present the performance of the fabricated PV cell after repeated cycles of ammonium nitrate dissolution and recrystallization. The self-healing is easier claimed than demonstrated. Recrystallization of ammonium nitrate crystals after temperature reduction during night-time is not guaranteed to occur along a thin film format. Natural crystallization can occur in the form of elongated and large crystals which will reduce the efficiency of dissolution after each cycle. The cycling performance is an essential requirement to satisfy the claim of self-healing.

Reviewer #3 (Remarks to the Author):

Title: Self-healing passive cooling utilizing endothermic reaction of $\text{NH}_4\text{NO}_3/\text{H}_2\text{O}$ driven by moisture sorption for photovoltaic cell

Comments:

A cooling method that combines latent cooling and endothermic reaction cooling is proposed to cool solar cells for PV applications. According to the manuscript, the average temperature of the PV cell can be reduced by $14.9\text{ }^\circ\text{C}$, corresponding to an efficiency improvement of 7.5%. In addition, the cost is also reduced by 5.1%. The new contribution and actual cooling effect of the proposed method should be re-checked. Furthermore, I do not think the impact warrants publication in Nature Communications. Please find my comments below.

1. While the combined cooling unit for PV cooling is a new structure, I question the performance and true impact of the proposed cooling method. The proposed cooling method may be general in the field of thermodynamics, what's the main highlight of this method when compared to other similar cooling methods, such as PCM cooling.

2. The experimental testing performed in the study is just a case study. Adsorption-desorption process and dissolution-crystallization process need different driven parameters (e.g., ambient temperature), so please explore the cooling performance under different weather conditions, such as in the winter season and summer seasons. Thus, the applicability of the combined cooling method can be proved.

3.The detailed analysis for cost is missing. It is very important to conduct a detailed cost analysis since the cost of PV applications is decreasing with a low price. So, the cost may determine the reasonability of the cooling method for industrial applications.

4.For equation (7), does heff consider the effect of radiation heat transfer? In addition, during the heat dissipation experiment, the apparatus should be placed outside so that radiation heat transfer between the cell and the cold sky occurs. Notably, radiative heat dissipation of the cell is also an important cooling method, which will be suppressed in the indoor environment.

5.Please explore the thickness of the zeolite layer on the cooling performance since a thicker zeolite layer provides more latent cooling ability but introduces additional heat resistance.

6.In Fig. 7, it seems that the testing configuration for natural and forced air cooling is quite different from that for latent cooling and endothermic reaction cooling, so the comparison of effective heat transfer coefficients is unfair. Besides, the temperature in Fig. 7c and d represents which components, cells, or other objects?

7.A detailed Reversible cycle testing (e.g., long time, more cycles, more operation conditions) can be added to highlight the feature of the proposed strategy.

8.For outdoor testing (i.e., Fig. 8), the testing details should be given. For example, how to deal with the generated electricity? Generally, a control with a maximal power point tracking function should be used to collect the transient generated power. If generated power is dissipated into the heat of the cell, the temperature of the cell will arise, which affects the accuracy of the monitored data and related conclusions.

9.In Fig. 8a, it is obvious that a hot spot is found on the center of the cell and the monitored temperature is overestimated, which directly affect the reported temperature reduction in conclusion. This phenomenon may be contributed by the setting of thermocouples, please check the experimental setting.

Title: Self-recovering passive cooling utilizing endothermic reaction of $\text{NH}_4\text{NO}_3/\text{H}_2\text{O}$ driven by water sorption for photovoltaic cell

Reviewers' comments

Reviewer #1

This manuscript reports a solar cell cooling strategy based on the endothermic reaction of $\text{NH}_4\text{NO}_3/\text{H}_2\text{O}$ driven by moisture sorption.

1. First of all, a very important and related reference [Photovoltaic panel cooling by atmospheric water sorption–evaporation cycle, *Nature Sustainability* 3, 636–643 (2020)] is missing. This work should be in the discussion of prior work. The novelty of the proposed approach, as compared to the existing work, should be clearly presented.

Answer: Thank you for your good comment. In INTRODUCTION, the literature survey on the passive cooling was revised based on the related references.

Before:

Photovoltaic (PV) cells convert energy from the sun directly to electricity, and are simple to set up because they have few components. The efficiency of a PV cell is approximately 10–25% considering that it can solely utilize a specific range of wavelengths⁸. The PV cell comprises a negative semiconductor with free electrons and positive semiconductors with holes. The wavelength that causes the photoelectric effect is different for each material, while materials with a large wavelength range have a low electric energy harvesting efficiency. Considering this, a tandem cell has been developed by stacking PV materials, which significantly increases the efficiency; however, it is difficult to utilize it because of its low economic feasibility⁹. Thin-film PV materials such as cadmium telluride, copper indium diselenide, and amorphous silicon, have been developed^{10, 11}. They are less efficient than thick films and are expensive or toxic. To date, building-integrated PV cells are mostly composed of monocrystalline silicon with high technical maturity, which presents high power efficiency¹².

The power efficiency of most PV devices is significantly affected by the cell temperature¹³. Specifically, the power efficiency of monocrystalline PV is reduced by 33.3% when the junction temperature is increased from 20 °C to 60 °C¹⁴. During electricity generation, the temperature rise owing to solar radiation is inevitable, and various cooling devices have been developed for stable power generation, including natural/forced air, hydraulic, heat pipes, phase change materials (PCMs), and thermoelectric cooling¹⁵. Natural air cooling is a phenomenon in which convective heat transfer occurs, owing to the density difference when the temperature of air rises around the cell. It is utilized for several PV cells because it does not require maintenance costs, although it has the least cooling effect. Forced air cooling generates

air velocity through a fan, which has a better cooling effect than natural convection¹⁶. However, when the ambient temperature is high, the cooling effect significantly decreases, and the maintenance cost is high as well. In addition, other challenges, such as noise, space efficiency, and energy consumption, make it less applicable. Hydraulic and thermoelectric cooling techniques, which respectively adopt working fluids and electricity as coolants, are not economically feasible¹⁷. PCMs absorb heat by phase change characteristics at a certain temperature, which effectively cools the PV cells. It is expensive and not utilized industrially¹⁸.

After (Page 3-4):

Solar energy is widely utilized to reduce the grid's energy load¹. Photovoltaic (PV) cells convert energy from the sun directly to electricity, and are simple to set up because they have few components. The efficiency of a PV cell is approximately 10–25% considering that it can solely utilize a specific range of wavelengths. To date, although highly efficient PV cells have been developed including the perovskite cell², building-integrated PV modules are mostly composed of monocrystalline silicon with high technical maturity³. On the other hand, the power efficiency of PV devices is significantly affected by the cell temperature⁴. For instance, the power efficiency of monocrystalline PV is reduced by 33.3% when the junction temperature is increased from 20 °C to 60 °C⁵. A greater reduction in efficiency by the temperature increase has also been reported for the perovskite cells⁶. During electricity generation, the temperature rise owing to solar radiation is inevitable, and various cooling devices have been developed for stable power generation including passive cooling and active cooling strategies^{7,8}.

Passive cooling methods generate cooling effect permanently without energy consumption⁹: Natural air cooling is a phenomenon in which convective heat transfer occurs, owing to the density difference when the temperature of air rises around the cell. Elminshawy et al.¹⁰ reduced the average working temperature of PV cell by 19% utilizing the natural air cooling with attached fins at a submerged area ratio of 20%. Natural liquid cooling composed of PV modules and a fluid storage tank was studied¹¹. The cooling effect of natural liquid cooling can be improved by using nanofluid as a coolant in microchannels¹². The phase change cooling technologies have been actively studied utilizing two-phase coolant with boiling temperature of 34 °C¹³, RT35¹⁴, Al₂O₃ composites¹⁵, and paraffin-based materials¹⁶. The heat pipe cooling device has been proposed and a maximum thermal efficiency can be improved by 14.2%¹⁷. Li et al.¹⁸ developed atmospheric water sorption-based cooling for PV cell, which can also harvest water during power generation. The radiant cooling methods have been also discussed¹⁹. Li et al. studied the nighttime radiative cooling for PV cell with water harvesting²⁰. Recently, co-generation of electricity and freshwater²¹, dynamic photovoltaic building envelopes²², and PV cooling-driven seawater desalination²³ were discussed as the PV cooling strategies.

Active cooling methods consume energy to achieve a cooling effect: Forced air cooling generates air velocity through a fan, which has a better cooling effect than the natural convection²⁴. However, when the ambient temperature is high, the cooling effect significantly decreases, and the maintenance cost becomes high as well⁹. In addition, other challenges, such as space efficiency and energy consumption, make it less applicable. Hydraulic and thermoelectric cooling techniques, which respectively adopt working fluids and electricity as coolants, are not economically feasible²⁵. On the other hand, PV thermal systems (PVTs) have been developed. As the fluid circulates at the bottom of PV cell through the channels, the thermal energy is harvested and PV cell cools down. PVTs are feasible in that the harvested

thermal energy can be utilized in buildings, although the energy is consumed to circulate fluid. The Al₂O₃ nanofluid²⁶, SiO₂ nanofluid²⁷, and microencapsulated PCM slurry fluid²⁸ were discussed as working fluids. Weinstein et al.²⁹ proposed beam splitting-based hybrid electric and thermal solar receiver.

- 1 Kim, S. *et al.* CO₂ capture-driven thermal battery using functionalized solvents for plus energy building application. *Energy Conversion and Management* **260**, 115606 (2022).
- 2 Wu, Y., Wang, Q., Chen, Y., Qiu, W. & Peng, Q. Stable perovskite solar cells with 25.17% efficiency enabled by improving crystallization and passivating defects synergistically. *Energy & Environmental Science* **15**, 4700-4709 (2022).
- 3 Zhu, L., Li, Q., Chen, M., Cao, K. & Sun, Y. A simplified mathematical model for power output predicting of Building Integrated Photovoltaic under partial shading conditions. *Energy Conversion and Management* **180**, 831-843 (2019).
- 4 Bezaatpour, J., Ghiasirad, H., Bezaatpour, M. & Ghaebi, H. Towards optimal design of photovoltaic/thermal facades: Module-based assessment of thermo-electrical performance, exergy efficiency and wind loads. *Applied Energy* **325**, 119785 (2022).
- 5 Chander, S., Purohit, A., Sharma, A., Nehra, S. & Dhaka, M. Impact of temperature on performance of series and parallel connected mono-crystalline silicon solar cells. *Energy Reports* **1**, 175-180 (2015).
- 6 Tress, W. *et al.* Performance of perovskite solar cells under simulated temperature-illumination real-world operating conditions. *Nature Energy* **4**, 568-574 (2019).
- 7 Zhao, B., Hu, M., Ao, X., Xuan, Q. & Pei, G. Spectrally selective approaches for passive cooling of solar cells: A review. *Applied Energy* **262**, 114548 (2020).
- 8 Verma, S., Mohapatra, S., Chowdhury, S., & Dwivedi, G. Cooling techniques of the PV module: A review. *Materials Today: Proceedings* **38**, 253-258 (2021)
- 9 Ghadikolaei, S. S. C. An enviroeconomic review of the solar PV cells cooling technology effect on the CO₂ emission reduction. *Solar Energy*, **216**, 468-492 (2021)
- 10 Elminshawy, N. A., El-Damhogi, D., Ibrahim, I., Elminshawy, A. & Osama, A. Assessment of floating photovoltaic productivity with fins-assisted passive cooling. *Applied Energy* **325**, 119810 (2022).
- 11 Maleki, A., Haghghi, A., Assad, M. E. H., Mahariq, I. & Nazari, M. A. A review on the approaches employed for cooling PV cells. *Solar Energy* **209**, 170-185 (2020).
- 12 Hussien, A. A., Abdullah, M. Z. & Moh'd A, A.-N. Single-phase heat transfer enhancement in micro/minichannels using nanofluids: theory and applications. *Applied Energy* **164**, 733-755 (2016).
- 13 Radwan, A., Ookawara, S. & Ahmed, M. Thermal management of concentrator photovoltaic systems using two-phase flow boiling in double-layer microchannel heat sinks. *Applied Energy* **241**, 404-419 (2019).
- 14 Chandel, S. & Agarwal, T. Review of cooling techniques using phase change materials for enhancing efficiency of photovoltaic power systems. *Renewable and Sustainable Energy Reviews* **73**, 1342-1351 (2017).
- 15 Salem, M., Elsayed, M., Abd-Elaziz, A. & Elshazly, K. Performance enhancement of the photovoltaic cells using Al₂O₃/PCM mixture and/or water cooling-techniques. *Renewable Energy* **138**, 876-890 (2019).
- 16 Fu, W. *et al.* High power and energy density dynamic phase change materials using pressure-enhanced close contact melting. *Nature Energy* **7**, 270-280 (2022).
- 17 Brahim, T. & Jemni, A. Parametric study of photovoltaic/thermal wickless heat pipe solar collector. *Energy Conversion and Management* **239**, 114236 (2021).
- 18 Li, R., Shi, Y., Wu, M., Hong, S. & Wang, P. Photovoltaic panel cooling by atmospheric water sorption–evaporation cycle. *Nature Sustainability* **3**, 636-643 (2020).

- 19 Li, W., Shi, Y., Chen, K., Zhu, L. & Fan, S. A comprehensive photonic approach for solar cell cooling. *ACS Photonics* **4**, 774-782 (2017).
- 20 Li, W. *et al.* Nighttime radiative cooling for water harvesting from solar panels. *ACS Photonics* **252**, 113432 (2019).
- 21 Ding, T. & Ho, G. W. Using the sun to co-generate electricity and freshwater. *Joule* **5**, 1639-1641 (2021).
- 22 Svetozarevic, B. *et al.* Dynamic photovoltaic building envelopes for adaptive energy and comfort management. *Nature Energy* **4**, 671-682 (2019).
- 23 Wang, W. *et al.* Integrated solar-driven PV cooling and seawater desalination with zero liquid discharge. *Joule* **5**, 1873-1887 (2021).
- 24 Lebbi, M. *et al.* Energy performance improvement of a new hybrid PV/T Bi-fluid system using active cooling and self-cleaning: Experimental study. *Applied Thermal Engineering* **182**, 116033 (2021).
- 25 Ilse, K. *et al.* Techno-economic assessment of soiling losses and mitigation strategies for solar power generation. *Joule* **3**, 2303-2321 (2019).
- 26 Kazemian, A., Khatibi, M., Maadi, S. R. & Ma, T. Performance optimization of a nanofluid-based photovoltaic thermal system integrated with nano-enhanced phase change material. *Applied Energy* **295**, 116859 (2021).
- 27 Hooshmandzade, N., Motevali, A., Seyedi, S. R. M. & Biparva, P. Influence of single and hybrid water-based nanofluids on performance of microgrid photovoltaic/thermal system. *Applied Energy* **304**, 117769 (2021).
- 28 Eisapour, M., Eisapour, A. H., Hosseini, M. & Talebizadehsardari, P. Exergy and energy analysis of wavy tubes photovoltaic-thermal systems using microencapsulated PCM nano-slurry coolant fluid. *Applied Energy* **266**, 114849 (2020).
- 29 Weinstein, L. A. *et al.* A hybrid electric and thermal solar receiver. *Joule* **2**, 962-975 (2018).

In addition, the novelty of the proposed approach was emphasized based on the literature review.

Before:

In summary, while low-cost cooling methods exhibit a low cooling effect, the methods with high cooling efficiencies are expensive. Herein, a novel type of passive cooling unit for a PV cell is proposed, and presents the best performance among cooling methods that have been reported in the literature^{19,20}. The method has no energy consumption and consists of cheap materials, making it highly applicable. The temperature of the cell decreases significantly during power generation by applying a passive cooling unit, and it can be repeatedly utilized, owing to its self-healing characteristics. The cooling mechanisms of the passive cooling unit have been investigated, and material design is possible for optimal cooling energy capacity, considering the operational conditions. Based on the feasibility study, it is economically feasible for industrial-scale applications with a highly efficient cooling performance.

After (Page 4):

To summarize, while low-cost cooling methods exhibit a low cooling effect, the methods with high cooling efficiencies are expensive **at the present level of technologies**. **Here**, a novel type of passive cooling unit for a PV cell is proposed to realize the best performance among the cooling methods that have been reported in the literature^{9,30}. **The cooling system**

induced by a chain reaction of latent cooling (water desorption process) and endothermic reaction (dissolution process) has not been attempted for PV applications, which is the main novelty of this study. The proposed method has no energy consumption and consists of cheap materials, making it highly applicable. Moreover, it can be repeatedly utilized, owing to its self-recovering characteristics. Meanwhile, a similar thermal management strategy for electronic devices was reported³¹. As heat is generated in electronic devices, the water molecules trapped in metal-organic frameworks (MOFs) are desorbed, which produces latent cooling effect. In this study, we have improved the atmospheric water desorption-driven cooling system by the chain reaction in which the desorbed water melts the crystal layer of ammonium nitrate and the secondary endothermic reaction cooling in a closed thin film.

- 30 Bijarniya, J. P., Sarkar, J. & Maiti, P. Review on passive daytime radiative cooling: Fundamentals, recent researches, challenges and opportunities. *Renewable and Sustainable Energy Reviews* **133**, 110263 (2020).
- 31 Wang, C. *et al.* A thermal management strategy for electronic devices based on moisture sorption-desorption processes. *Joule* **4**, 435-447 (2020).

2. In addition, in the introduction discussion of the existing approach, references of radiative cooling are also missing, for example [A Comprehensive Photonic Approach for Solar Cell Cooling, *ACS Photonics* **4**, 774–782 (2017)]. In addition, the use of solar cell cooling at night for water sorption has also been proposed [Nighttime Radiative Cooling for Water Harvesting from Solar Panels, *ACS Photonics* **8**, 269–275 (2021)], which is also related to the proposed approach.

Answer: Based on your comments, the discussion of the existing approach was improved by adding related studies including:

- 18 Li, R., Shi, Y., Wu, M., Hong, S. & Wang, P. Photovoltaic panel cooling by atmospheric water sorption–evaporation cycle. *Nature Sustainability* **3**, 636-643 (2020).
- 19 Li, W., Shi, Y., Chen, K., Zhu, L. & Fan, S. A comprehensive photonic approach for solar cell cooling. *ACS Photonics* **4**, 774-782 (2017).
- 20 Li, W. *et al.* Nighttime radiative cooling for water harvesting from solar panels. *ACS Photonics* **252**, 113432 (2019).

3. The authors claim “Herein, a novel type of passive cooling unit for a PV cell is proposed, and presents the best performance among cooling methods that have been reported in the literature^{19,20}”. However, in the experiments part, the main comparison is with natural air cooling. A comparison of the proposed approach with all existing approaches should be presented, to support their claim. Also, references 19, 20 do not serve as supporting references in this context.

Answer: Thank you for the good point. The comparison of water desorption-driven endothermic reaction (WD-ER) cooling unit with other technologies was performed in Table 1. The cooling energy density of WD-ER reaches 2,876 kJ/kg (PCMs present 100-300 kJ/kg and moisture sorption-driven cooling presents 1,950 kJ/kg (Wang et al. *Joule* 2020)). The cooling power of WD-ER is 403 W/m², which outperforms the radiative cooling methods of 40.1 W/m² (Raman et al. *Nature* 2020) and 310 W/m² (Wang et al. *Joule* 2020).

Table 1 compares the cooling performance of WD-ER unit with other methods reported in the literature^{18,31,40-51}. The cooling energy density of PCMs ranges 100-300 kJ/kg. Although the temperature of PCMs is maintained at the melting point during the cooling process, the sensible heat change of the PV cell occurs continuously (i.e., the temperature of PV cell is not kept constant regardless of the state of PCMs). Moreover, the cooling energy density of MIL-101(Cr) coating layer is high at 1,950 kJ/kg using ambient moisture sorption-desorption process for electronic device applications³¹. Similarly, the PV cooling by atmospheric water sorption-evaporation cycle is proposed, and the cooling power is 295 W/m² with temperature drop of 10 °C. The radiative methods present cooling power of 40-300 W/m². The WD-ER of the present study outperforms the literature cooling methods in terms of cooling energy density (2,876 kJ/kg) and cooling power (403 W/m²).

Table 1. Comparison of WD-ER cooling unit with passive cooling methods

Methods		Density (kJ/kg)	Conditions and remarks	References
WD-ER	13X/H ₂ O/NH ₄ NO ₃	2,876	Operating range: 20-70 °C Cooling power: 403 W/m ² Temperature drop: 15.1 °C	
Phase change materials	RT55	170	Melting: 51-57 °C	40
	RT28HC	220	Melting: 28 °C	41
	C-PCM	199	Melting: 21.8 °C	42
	OM47	196	Melting: 48 °C	43
	MF-PW30	139.8	Melting: 56.8 °C crystallization: 45.1 °C	44
	PCMB	141.7	Steady range: 40.4-84.9 °C	45
	GO-EHS/PAAAM	200.3	Melting: 23 °C	46
	Eicosane	237.4	Melting: 36.5 °C	47
	Tricosane	269.2	Melting: 42-48 °C	48
	MIL-101(Cr)	1,950	Operating range: 20-70 °C	31
PAN-CNT-CaCl ₂	-	Operating range: 20-70 °C Cooling power: 295 W/m ² Temperature drop: 10.0 °C	18	
Radiative cooling	HfO ₂ and SiO ₂	-	Operating range: 22 °C Cooling power: 40.1 W/m ²	49
	Cooling assembly system	-	Operating range: 74 °C Cooling power: 310 W/m ² Temperature drop: 36.6 °C	50
	Silica pyramid	-	Operating range: 20-70 °C Temperature drop: 18.3 °C	51

- 40 Caliskan, H., Gurbuz, H., Sohret, Y. & Ates, D. Thermal analysis and assessment of phase change material utilization for heating applications in buildings: A modelling. *Journal of Energy Storage* **50**, 104593 (2022).
- 41 Zhao, Y. *et al.* Expanded graphite–Paraffin composite phase change materials: Effect of particle size on the composite structure and properties. *Applied Thermal Engineering* **171**, 115015 (2020).
- 42 Velmurugan, K. *et al.* Experimental studies on photovoltaic module temperature

- reduction using eutectic cold phase change material. *Solar Energy* **209**, 302-315 (2020).
- 43 Malvika, A., Arunachala, U. & Varun, K. Sustainable passive cooling strategy for photovoltaic module using burlap fabric-gravity assisted flow: A comparative Energy, exergy, economic, and enviroeconomic analysis. *Applied Energy* **326**, 120036 (2022).
- 44 Jing, J. H. *et al.* Melamine foam-supported form-stable phase change materials with simultaneous thermal energy storage and shape memory properties for thermal management of electronic devices. *ACS Applied Materials & Interfaces* **11**, 19252-19259 (2019).
- 45 Wu, W. *et al.* Preparation and thermal conductivity enhancement of composite phase change materials for electronic thermal management. *Energy Conversion and Management* **101**, 278-284 (2015).
- 46 Liu, Y., Yang, Y. & Li, S. Graphene oxide modified hydrate salt hydrogels: form-stable phase change materials for smart thermal management. *Journal of Materials Chemistry A* **4**, 18134-18143 (2016).
- 47 Baby, R. & Balaji, C. Experimental investigations on phase change material based finned heat sinks for electronic equipment cooling. *International Journal of Heat and Mass Transfer* **55**, 1642-1649 (2012).
- 48 Weng, Y. C., Cho, H. P., Chang, C. C. & Chen, S. L. Heat pipe with PCM for electronic cooling. *Applied Energy* **88**, 1825-1833 (2011).
- 49 Raman, A. P., Anoma, M. A., Zhu, L., Rephaeli, E. & Fan, S. Passive radiative cooling below ambient air temperature under direct sunlight. *Nature* **515**, 540-544 (2014).
- 50 Wang, Z. *et al.* Lightweight, passive radiative cooling to enhance concentrating photovoltaics. *Joule* **4**, 2702-2717 (2020).
- 51 Zhu, L., Raman, A., Wang, K. X., Abou Anoma, M. & Fan, S. Radiative cooling of solar cells. *Optica* **1**, 32-38 (2014).

4. The writing and organization of this manuscript should also be significantly improved. The first two paragraphs are too generic. They should be more related to the cooling of solar cells. Too many figures in the main text. Many figure data in the main text are just supporting materials characterization data. They did not serve as direct proof of the main claim. They should be reorganized or moved to supporting information.

Answer: In INTRODUCTION, the first two paragraphs were revised to focus on the cooling of solar cells as follows;

Before:

Owing to the scientific revolution, CO₂ emissions and energy consumption are mainly discussed as challenges. In particular, while it may differ from the environment^{1, 2}, the energy consumption of buildings is immense, which is approximately 30%–50% of the total energy consumption. Zero-energy buildings have been developed by generating renewable energy, and reducing the energy load to improve the energy consumption pattern of buildings³. Specifically, wind energy is limited in cities, owing to noise and safety issues⁴, and hydropower, marine power, and waste fuels are difficult for small system applications⁵. Ground source heat pumps are currently being investigated to improve the coefficient of performance, and they cannot be independently operated from the energy grid⁶. Solar energy is widely utilized and can be harvested in the form of thermal and electric energy. Solar thermal collectors are highly efficient; however, they have disadvantages, in that thermal energy is transferred through working fluids such as water and engine oil, which requires several components for building

applications⁷.

Photovoltaic (PV) cells convert energy from the sun directly to electricity, and are simple to set up because they have few components. The efficiency of a PV cell is approximately 10–25% considering that it can solely utilize a specific range of wavelengths⁸. The PV cell comprises a negative semiconductor with free electrons and positive semiconductors with holes. The wavelength that causes the photoelectric effect is different for each material, while materials with a large wavelength range have a low electric energy harvesting efficiency. Considering this, a tandem cell has been developed by stacking PV materials, which significantly increases the efficiency; however, it is difficult to utilize it because of its low economic feasibility⁹. Thin-film PV materials such as cadmium telluride, copper indium diselenide, and amorphous silicon, have been developed^{10, 11}. They are less efficient than thick films and are expensive or toxic. To date, building-integrated PV cells are mostly composed of monocrystalline silicon with high technical maturity, which presents high power efficiency¹².

After (Page 3)

Solar energy is widely utilized to reduce the grid's energy load¹. Photovoltaic (PV) cells convert energy from the sun directly to electricity, and are simple to set up because they have few components. The efficiency of a PV cell is approximately 10–25% considering that it can solely utilize a specific range of wavelengths. To date, although highly efficient PV cells have been developed including the perovskite cell², building-integrated PV modules are mostly composed of monocrystalline silicon with high technical maturity³. On the other hand, the power efficiency of PV devices is significantly affected by the cell temperature⁴. For instance, the power efficiency of monocrystalline PV is reduced by 33.3% when the junction temperature is increased from 20 °C to 60 °C⁵. A greater reduction in efficiency by the temperature increase has also been reported for the perovskite cells⁶. During electricity generation, the temperature rise owing to solar radiation is inevitable, and various cooling devices have been developed for stable power generation including passive cooling and active cooling strategies^{7,8}.

Passive cooling methods generate cooling effect permanently without energy consumption⁹: Natural air cooling is a phenomenon in which convective heat transfer occurs, owing to the density difference when the temperature of air rises around the cell. Elminshawy et al.¹⁰ reduced the average working temperature of PV cell by 19% utilizing the natural air cooling with attached fins at a submerged area ratio of 20%. Natural liquid cooling composed of PV modules and a fluid storage tank was studied¹¹. The cooling effect of natural liquid cooling can be improved by using nanofluid as a coolant in microchannels¹². The phase change cooling technologies have been actively studied utilizing two-phase coolant with boiling temperature of 34 °C¹³, RT35¹⁴, Al₂O₃ composites¹⁵, and paraffin-based materials¹⁶. The heat pipe cooling device has been proposed and a maximum thermal efficiency can be improved by 14.2%¹⁷. Li et al.¹⁸ developed atmospheric water sorption-based cooling for PV cell, which can also harvest water during power generation. The radiant cooling methods have been also discussed¹⁹. Li et al. studied the nighttime radiative cooling for PV cell with water harvesting²⁰. Recently, co-generation of electricity and freshwater²¹, dynamic photovoltaic building envelopes²², and PV cooling-driven seawater desalination²³ were discussed as the PV cooling strategies.

Active cooling methods consume energy to achieve a cooling effect: Forced air cooling generates air velocity through a fan, which has a better cooling effect than the natural

convection²⁴. However, when the ambient temperature is high, the cooling effect significantly decreases, and the maintenance cost becomes high as well⁹. In addition, other challenges, such as space efficiency and energy consumption, make it less applicable. Hydraulic and thermoelectric cooling techniques, which respectively adopt working fluids and electricity as coolants, are not economically feasible²⁵. On the other hand, PV thermal systems (PVTs) have been developed. As the fluid circulates at the bottom of PV cell through the channels, the thermal energy is harvested and PV cell cools down. PVTs are feasible in that the harvested thermal energy can be utilized in buildings, although the energy is consumed to circulate fluid. The Al₂O₃ nanofluid²⁶, SiO₂ nanofluid²⁷, and microencapsulated PCM slurry fluid²⁸ were discussed as working fluids. Weinstein et al.²⁹ proposed beam splitting-based hybrid electric and thermal solar receiver.

In addition, the table of contents was revised in the similar format as the latest papers published in “Nature Communications”.

Before:

1. INTRODUCTION
2. METHODS
 - 2.1. Strategy
 - 2.2. Materials preparation
 - 2.3. Characterization
 - 2.4. Theory of endothermic reaction cooling
 - 2.5. Heat dissipation experiments
 - 2.6. Industrial applications
3. RESULTS AND DISCUSSION
 - 3.1. Mesoporous materials for latent cooling
 - 3.2. Solvent pairs for endothermic reaction cooling
 - 3.3. Heat dissipation performance
 - 3.4. Industrial applications of MS-ER
4. CONCLUSIONS

After:

INTRODUCTION

RESULTS AND DISCUSSION

Working principle of self-recovering passive cooling

Porous materials for latent cooling

Solvent pairs for endothermic reaction cooling

Lab-scale heat dissipation performance using WD-ER

Practical applications of WD-ER for photovoltaic cell cooling

CONCLUSIONS

METHODS

Some figures related with characterization of materials were moved to Supporting information.

Other figures were reorganized/supplemented to focus on direct proof of the main claim.

Before:

Figure 1. Schematic diagram of passive cooling unit using endothermic reaction of ammonium nitrate and water driven by water sorption-desorption for PV cell applications; a) Passive cooling unit integrated with PV cell; b) Cooling effect during power generation in response to solar radiation; c) Cooling principle considering internal composition; d) Self-healing process during the night.

Figure 2. Manufacturing method of PV cell with MS-ER cooling unit; a) PV cell composed of monocrystalline silicon; b) Waterproofing coating to protect power terminal unit; c) water-saturated zeolite 13X coating for latent cooling; d) Ammonium nitrate coating for endothermic reaction cooling; e) Sealing unit for safe operation.

Figure 3. Characterization of mesoporous materials; a) N_2 adsorption curve; b) XRD pattern; c) FE-SEM images; d) micropore size distribution; e) mesopore size distribution.

Figure 4. Water sorption performance of mesoporous materials; a) Zeolite 13X; b) Zeolite 5A; c) Zeolite 3A; d) Zeolite Y; e) SAPO-34; f) SSZ-13.

Figure 5. Latent cooling behavior of mesoporous materials; a) Reaction heat versus water loading; b) Latent cooling energy density considering water sorption capacity and heat of reaction.

Figure 6. Theoretical analysis of endothermic reaction pairs for MS-ER cooling; a) Ideal solubility and heat of reaction; b) Endothermic reaction cooling energy density.

Figure 7. Heat dissipation performance in terms of cooling methods; a) Natural air cooling; b) Forced air cooling; c) Latent cooling; d) Endothermic reaction cooling; e) MS-ER cooling; f) Repeated MS-ER cooling.

Figure 8. Application of MS-ER unit for PV power generation; a) Schematic of PV cell with MS-ER unit; b) Temperature variation of PV cell during a day; c) Net absorption heat density of MS-ER unit without natural convection; d) Current-voltage characteristics in terms of temperature; e) Cost analysis and power density for long-term use.

Figure S1. Visualization analysis during heat dissipation process; a) Schematic of thermal imaging experiment; b) Latent cooling using zeolite 13X/ H_2O ; c) Endothermic reaction cooling using NH_4NO_3/H_2O

Figure S2. Experimental device for evaluation of heat dissipation performances

Figure S3. FE-SEM images for mesoporous materials; a) Zeolite 13X; b) Zeolite 5A; c) Zeolite 3A; d) Zeolite Y; e) SAPO-34; f) SSZ-13

After:

Figure 1. Schematic diagram of **self-recovering** passive cooling unit **utilizing the chain reaction of water desorption (primary latent cooling) and dissolution of ammonium nitrate in water (secondary endothermic reaction cooling)** for PV cell applications; (a) Passive cooling unit integrated with PV cell; (b) Cooling **power** during electricity generation in response to solar radiation; (c) Cooling principle considering internal composition during the day; (d) Self-recovering process during the night.

Figure 2. Water sorption performance of porous materials; (a) Zeolite 13X; (b) Zeolite 5A; (c) Zeolite 3A; (d) Zeolite Y; (e) SAPO-34; (f) SSZ-13.

Figure 3. Latent cooling behavior of **porous** materials; (a) Reaction heat versus water loading;

(b) Latent cooling energy density considering water sorption capacity and heat of reaction.

Figure 4. Theoretical analysis of endothermic reaction pairs for WD-ER cooling; (a) Ideal solubility and heat of reaction; (b) Endothermic reaction cooling energy density.

Figure 5. Heat dissipation performance in terms of cooling methods; (a) Natural air cooling; (b) Forced air cooling; (c) Latent cooling; (d) Endothermic reaction cooling; (e) WD-ER cooling; (f) Repeated WD-ER cooling.

Figure 6. Heat dissipation performance of latent cooling in terms of zeolite film thickness

Figure 7. Application of WD-ER unit for PV power generation; (a) Schematic of PV cell with WD-ER unit; (b) Temperature variation of PV cell during a day; (c) Net cooling power density of WD-ER unit without natural convection.

Figure 8. Practical working performance of WD-ER unit; (a) Cooling power density and average temperature variation during the day; (b) Current-voltage characteristics in terms of temperature.

Figure S1. Characterization of porous materials; (a) N₂ adsorption curve; (b) XRD pattern; (c) Micropore size distribution; (d) Mesopore size distribution.

Figure S2. FE-SEM images for porous materials; (a) Zeolite 13X; (b) Zeolite 5A; (c) Zeolite 3A; (d) Zeolite Y; (e) SAPO-34; (f) SSZ-13

Figure S3. Visualization analysis during heat dissipation process; (a) Schematic of thermal imaging experiment; (b) Latent cooling using zeolite 13X/H₂O; (c) Endothermic reaction cooling using NH₄NO₃/H₂O

Figure S4. Experimental device for evaluation of heat dissipation performances

Figure S5. Manufacturing method of PV cell with WD-ER cooling unit; (a) PV cell composed of monocrystalline silicon; (b) Waterproofing coating to protect power terminal unit; (c) water-saturated zeolite 13X coating for latent cooling; (d) Ammonium nitrate coating for endothermic reaction cooling; (e) Sealing unit for safe operation.

5. While too many supporting data are presented in the manuscript, the key data is insufficient. For example, the cooling effect of this approach (Figure 8b) should have at least shown a few full-day cycles to show it can self-sustainably work with the daytime and nighttime working mechanisms.

Answer: Most of the supporting data (characterization data) was moved to Supplementary information. Moreover, based on your comments, the key data related to the performance of WD-ER cooling unit was supplemented.

The full-day cycle performance of WD-ER unit was evaluated when the outdoor temperature is 30.5-31.7 °C (Page 15-16), which verify the self-recovering characteristics.

The cyclic cooling performance of WD-ER is evaluated in Figure 8(a), when the outdoor temperature is 30.5-31.7 °C. During the power generation period, the temperature of PV cell with WD-ER reaches a maximum of 60 °C, and after the sunset, the temperature drops to around 25 °C, confirming the self-recovering characteristics (a temperature difference of 35 °C, which is maintained for more than 8 hours, is sufficient for recovering process. In addition, the radiation cooling effect is also significant at night.). The average cooling power density is uniform at 375-419 W/m², although it slightly differs in terms of external conditions. In winter, the cooling effect of the WD-ER unit could be more important as the working temperature range becomes larger (In general, the temperature range of PV cell is 0-55 °C during power generation, and at night, the temperature drops to -10 °C). Figure 8(b) illustrates the current-

voltage characteristics of the monocrystalline silicon. As the temperature rises, the open-circuit voltage increases, while the short-circuit current decreases. The maximum power is inversely proportional to temperature (25 °C: 91.3 mW → 60 °C: 81.4 mW). In summary, the power efficiency of PV cell with WD-ER unit is improved by approximately 10% compared to that of PV cell with natural air cooling. The improvement of power efficiency has been analyzed based on a commercially available monocrystalline silicon.

Figure 8. Practical working performance of WD-ER unit; (a) Cooling power density and average temperature variation during the day; (b) Current-voltage characteristics in terms of temperature.

The physical working principle of WD-ER was revised to focus on key conclusions (Page 5-6).

The deeper reflection on the physical working principle of WD-ER unit is described. The water desorption process from zeolite 13X is expressed as Eq. (1), where Q_d is desorption energy (kJ), m_a is adsorbent mass (kg), h_d is average desorption enthalpy of water (kJ/kg-H₂O), and x_w is adsorption capacity (kg-H₂O/kg-13X). The subscripts i , and f represent the initial and final states, respectively.

$$Q_d = m_a \cdot h_d \cdot (x_{w,i} - x_{w,f}) \quad (1)$$

In the heat exchange process between the PV cell and the thin film (WD-ER unit), the Biot number is less than 0.01, which supports the lumped system analysis in the longitudinal direction, and most of the radiant energy is consumed for water desorption. Moreover, the solar thermal energy is uniformly supplied to the PV modules, enabling one-dimensional analysis. The well-known diffusion equation of desorbed water is Eq. (2)³¹, where m_w is desorbed water mass (kg), K^* is modified mass transfer coefficient (m/s), A is cross-sectional area (m²), C_s is concentration of water on the particle surface (kg/m³), and C_f is free stream concentration (kg/m³), respectively (This equation is defined as a pseudo first-order kinetic model³⁵).

$$\frac{d(m_w)}{dt} = K^* \cdot A \cdot (C_s - C_f) \quad (2)$$

However, in the case of WD-ER unit, the diffusion of water molecules is limited in that the concentration difference is not significant. Although the desorption enthalpy of water from microporous materials is lower than that of water vaporization, the cooling effect is estimated as high as 27.4 kJ/mol-H₂O.

The endothermic reaction cooling of dissolution process is estimated by Eq. (3), where Q_e is endothermic reaction energy (kJ) and x_s is adsorption capacity (kg-NH₄NO₃/kg-H₂O). The subscript s represent solute.

$$Q_e = m_w \cdot h_e \cdot (x_{s,f} - x_{s,i}) \quad (3)$$

The dissolution enthalpy (h_e ; kJ/kg-solute) is the difference between the formation enthalpies of the aqueous and solid phases. The ideal solubility is defined as the saturated mole fraction of the solute in the solution. It is evaluated by the equilibrium of the chemical potential between the solid and solution states as Eq. (4), where y is the mole fraction of solute in the solution, R is the gas constant, T is temperature, and T_F is the freezing temperature of the solute, respectively.

$$\ln(y) = \frac{h_e}{R} \left(\frac{1}{T} - \frac{1}{T_F} \right) \quad (4)$$

The total cooling energy density of WD-ER unit can be obtained by considering the chain reaction of water desorption and solute dissolution.

The effective heat transfer coefficient (W/m²·K) during the heat dissipation process was discussed additionally. The effect of thickness variation of zeolite 13X layer on the cooling performance was evaluated.

Before:

As presented in Figure 1, the cooling performance of MS-ER can be maximized by a combined structure comprising a latent cooling layer of mesoporous materials and an endothermic reaction cooling layer of ammonium nitrate crystals that is driven by water as a solvent, which generates a chain reaction of cooling to enhance the cooling energy density. The heat dissipation performance of the MS-ER cooling unit is illustrated in Figure 7e. The temperature gradient of MS-ER cooling unit is the lowest compared to other cooling methods, and the average effective heat transfer coefficient is 64.1 W/m²·K. A sudden temperature drop at 45 °C is also observed, as in the case of the NH₄NO₃/H₂O pair. The mass of the solvent required for operation can be estimated using Eq. (11), considering the ideal solubility and dissolution energy of NH₄NO₃ and the desorption energy of water. Because the cooling of MS-ER is a two-step process of water desorption from zeolite 13X and dissolution of NH₄NO₃ in water, the working mass of the solvent is significantly less than the mass required for the latent cooling or endothermic reaction cooling method.

After (Page 13-14):

To further discuss the heat dissipation mechanism, the average effective heat transfer coefficient of the WD-ER unit (13X/H₂O/NH₄NO₃) is 64.1 W/m²·K, far superior to the sum of effective heat transfer coefficients of latent heat cooling utilizing 13X/H₂O (29 W/m²·K) and endothermic cooling utilizing NH₄NO₃/H₂O (19.5 W/m²·K). The latent cooling mechanism of WD-ER is invariant compared to that of 13X/H₂O in that the thermal energy supplied to the PV is transferred to the thin film and it induces water desorption. However, the WD-ER

presents a different aspect in terms of dissolution process when only endothermic reaction cooling ($\text{NH}_4\text{NO}_3/\text{H}_2\text{O}$) is performed. As thermal energy is directly supplied to $\text{NH}_4\text{NO}_3/\text{H}_2\text{O}$, the solubility increases and NH_4NO_3 is dissolved, resulting in an endothermic reaction. In this process, the reaction area between NH_4NO_3 and H_2O is limited in that water cannot flow between the NH_4NO_3 crystal layers. When endothermic cooling is induced after the latent cooling for WD-ER, water molecules generated in the form of droplets permeate between the NH_4NO_3 crystal layers and the reaction area widens (in Figure S3, the water droplets are generated on the surface of zeolite 13X). In summary, the cooling power of WD-ER increases due to the improvement of dissolution process by widening the interfacial area. On the other hand, the cooling energy density of WD-ER (2,876 kJ/kg- H_2O) is similar to the sum of cooling energy densities of latent cooling (346.6 kJ/kg-13X is equivalent to 1,815 kJ/kg- H_2O) and endothermic reaction cooling (1,386 kJ/kg- H_2O) with a 10% error.

It is evaluated that after the **WD-ER** cooling unit performs heat dissipation, whether all processes are self-recovered by cyclic experiments. In Figure 5(f), as the effective heat transfer coefficient is maintained at 64.1–68.9 $\text{W}/\text{m}^2\cdot\text{K}$ in five cycles, it is concluded that the WD-ER cooling unit exhibits self-recovering characteristics. Specifically, at 20 °C, zeolite 13X completely adsorbs water again, NH_4NO_3 crystallizes to form a hard layer, and **WD-ER** exhibits a solid-like structure. When heat is supplied, water is regenerated from zeolite 13X, and the water dissolves NH_4NO_3 again to generate a cooling effect. Accordingly, at 50 °C, an aqueous solution in which NH_4^+ and NO_3^- are dissolved is observed, and the color of zeolite 13X is pale, indicating that water is desorbed. **Note that, in the solidification process of NH_4NO_3 , crystals could be randomly formed. However, the reproducibility problem due to the random crystallization does not occur in Figure 5(f) because the interfacial area of dissolution is large as the water droplets desorbed from zeolite 13X permeate between them. Moreover, the thickness change of zeolite 13X film layer does not affect the heat dissipation kinetics (rate/behavior), and only the cooling energy density is varied as presented in Figure 6. As the temperature change is the driving force of water desorption, a certain amount of water molecules are regenerated regardless of the thickness of the layer (The rate-limiting step of water desorption process is dominated by the cross-sectional area instead of the thickness)³¹. As the thickness of zeolite 13X film is in the range of 1-5 mm, the thermal resistance can be neglected with the Biot number of 1.5-7.5 10^{-5} .**

Figure 6. Heat dissipation performance of latent cooling in terms of zeolite film thickness

The cooling power density (W/m^2) of WD-ER unit was supplemented in practical applications (Page 15).

Figure 7(c) illustrates the net cooling power density considering the temperature difference between natural air and WD-ER cooling (temperature data were converted into cooling energy per unit area of PV cell utilizing the smoothing filter of Savitzky-Golay theory. After dividing it by the measurement time interval, the cooling power density was obtained). When the cell temperature is in the range of 20–45 °C (time range of 0–3,500 seconds), the radiation heat is absorbed at $380 \text{ W}/\text{m}^2$, which corresponds to the desorption energy of water from zeolite 13X (latent cooling). The absorption heat amount rapidly increases to $495 \text{ W}/\text{m}^2$ at 45 °C as the NH_4NO_3 layer reacts with solvent, which is the endothermic reaction cooling for 3,500–4,000 seconds. It is similar to the case when WD-ER cooling unit is applied for lab-scale heat dissipation experiment in Figure 5(e), the temperature rapidly decreases around 45 °C. Further, the cooling power density is maintained at $431 \text{ W}/\text{m}^2$ with the combined water desorption and solute dissolution. When the cooling energy density reaches 2,500 kJ/kg at approximately 9,000 seconds, the cooling performance is significantly reduced as the WD-ER cooling unit begins to be saturated. Finally, the cooling energy density is 2,876 kJ/kg, which is sufficient for utilization under high ambient temperature conditions.

Figure 7. Application of WD-ER unit for PV power generation; (c) Net cooling power density of WD-ER unit without natural convection.

Finally, the cooling performance of WD-ER unit outperforms that of other methods.

Table 1 compares the cooling performance of WD-ER unit with other methods reported in the literature^{18,31,40-51}. The cooling energy density of PCMs ranges 100–300 kJ/kg. Although the temperature of PCMs is maintained at the melting point during the cooling process, the sensible heat change of the PV cell occurs continuously (i.e., the temperature of PV cell is not kept constant regardless of the state of PCMs). Moreover, the cooling energy density of MIL-101(Cr) coating layer is high at 1,950 kJ/kg using ambient moisture sorption-desorption process for electronic device applications³¹. Similarly, the PV cooling by atmospheric water sorption-evaporation cycle is proposed, and the cooling power is $295 \text{ W}/\text{m}^2$ with temperature drop of 10 °C. The radiative methods present cooling power of 40–300 W/m^2 . The WD-ER of the present study outperforms the literature cooling methods in terms of cooling energy density (2,876 kJ/kg) and cooling power ($403 \text{ W}/\text{m}^2$).

Table 1. Comparison of WD-ER cooling unit with passive cooling methods

Methods		Density (kJ/kg)	Conditions and remarks	References
WD-ER	13X/H ₂ O/NH ₄ NO ₃	2,876	Operating range: 20-70 °C Cooling power: 403 W/m ² Temperature drop: 15.1 °C	
Phase change materials	RT55	170	Melting: 51-57 °C	40
	RT28HC	220	Melting: 28 °C	41
	C-PCM	199	Melting: 21.8 °C	42
	OM47	196	Melting: 48 °C	43
	MF-PW30	139.8	Melting: 56.8 °C crystallization: 45.1 °C	44
	PCMB	141.7	Steady range: 40.4-84.9 °C	45
	GO-EHS/PAAAM	200.3	Melting: 23 °C	46
	Eicosane	237.4	Melting: 36.5 °C	47
	Tricosane	269.2	Melting: 42-48 °C	48
	MIL-101(Cr)	1,950	Operating range: 20-70 °C	31
PAN-CNT-CaCl ₂	-	Operating range: 20-70 °C Cooling power: 295 W/m ² Temperature drop: 10.0 °C	18	
Radiative cooling	HfO ₂ and SiO ₂	-	Operating range: 22 °C Cooling power: 40.1 W/m ²	49
	Cooling assembly system	-	Operating range: 74 °C Cooling power: 310 W/m ² Temperature drop: 36.6 °C	50
	Silica pyramid	-	Operating range: 20-70 °C Temperature drop: 18.3 °C	51

Reviewer #2

The manuscript elaborates the use of moisture sorption driven endothermic reaction (MS-ER)-based cooling of the photovoltaic cells for reduction in photovoltaic power generation costs over a 10-year period and increase in energy production density.

The MS-ER cooling is achieved using a water saturated zeolite film and an ammonium nitrate film under the photovoltaic cell. The increase in PV cell temperature leads to water desorption from the zeolite which results in latent cooling. The expunged water vapours dissolve ammonium nitrate crystals which results in further endothermic cooling. When the PV cell cools at night it was claimed that the reduction in temperature leads to precipitation of ammonium nitrate crystals, and water is adsorbed by the zeolite layer. This is the claimed self-healing effect.

Overall, the manuscript presents good scientific discussion, good figures, logical development of arguments, good combination of experimental and numerical approaches and commendable novelty in the use of porous materials as water reservoirs. However, the research still lacks critical elements essential for qualifying into an esteemed science journal such as Nature Communications.

Answer: Thank you for your thoughtful comments and efforts in evaluating the manuscript. Based on the advice, the manuscript was faithfully/thoroughly revised. The contents of the manuscript have been rearranged to suit “Nature Communications”.

These issues are described below:

1) The concept of photovoltaic cell cooling for improvement in cell efficiency is now well proven. Different approaches have been demonstrated for reduction of PV cell temperature such as air cooling, use of cooling fins, use of microchannels, recirculating fluid cooling (water/oil/others), cooling induced by water spray, cooling induced by Peltier cells and cooling induced by the use of phase change materials. For instance, reviews of such methods are available in the literature.

(<https://doi.org/10.1016/j.matpr.2020.07.130>, <https://doi.org/10.1016/j.solener.2021.01.016>)

Answer: Thank you for your good comment. In INTRODUCTION, the different approaches of PV cooling including passive/active technologies was discussed based on the suggested references.

Before:

Owing to the scientific revolution, CO₂ emissions and energy consumption are mainly discussed as challenges. In particular, while it may differ from the environment^{1, 2}, the energy consumption of buildings is immense, which is approximately 30%–50% of the total energy consumption. Zero-energy buildings have been developed by generating renewable energy, and reducing the energy load to improve the energy consumption pattern of buildings³. Specifically, wind energy is limited in cities, owing to noise and safety issues⁴, and hydropower, marine power, and waste fuels are difficult for small system applications⁵. Ground source heat pumps are currently being investigated to improve the coefficient of performance, and they cannot be independently operated from the energy grid⁶. Solar energy is widely utilized and can be harvested in the form of thermal and electric energy. Solar thermal collectors are highly efficient; however, they have disadvantages, in that thermal energy is transferred through working fluids such as water and engine oil, which requires several components for building applications⁷.

Photovoltaic (PV) cells convert energy from the sun directly to electricity, and are simple to set up because they have few components. The efficiency of a PV cell is approximately 10–25% considering that it can solely utilize a specific range of wavelengths⁸. The PV cell comprises a negative semiconductor with free electrons and positive semiconductors with holes. The wavelength that causes the photoelectric effect is different for each material, while materials with a large wavelength range have a low electric energy harvesting efficiency. Considering this, a tandem cell has been developed by stacking PV materials, which significantly increases the efficiency; however, it is difficult to utilize it because of its low economic feasibility⁹. Thin-film PV materials such as cadmium telluride, copper indium diselenide, and amorphous silicon, have been developed^{10, 11}. They are less efficient than thick films and are expensive or toxic. To date, building-integrated PV cells are mostly composed of monocrystalline silicon with high technical maturity, which presents high power efficiency¹².

The power efficiency of most PV devices is significantly affected by the cell temperature¹³. Specifically, the power efficiency of monocrystalline PV is reduced by 33.3% when the junction temperature is increased from 20 °C to 60 °C¹⁴. During electricity generation, the temperature rise owing to solar radiation is inevitable, and various cooling devices have been developed for stable power generation, including natural/forced air, hydraulic, heat pipes, phase change materials (PCMs), and thermoelectric cooling¹⁵. Natural air cooling is a phenomenon in which convective heat transfer occurs, owing to the density difference when the temperature of air rises around the cell. It is utilized for several PV cells because it does not require maintenance costs, although it has the least cooling effect. Forced air cooling generates air velocity through a fan, which has a better cooling effect than natural convection¹⁶. However, when the ambient temperature is high, the cooling effect significantly decreases, and the maintenance cost is high as well. In addition, other challenges, such as noise, space efficiency, and energy consumption, make it less applicable. Hydraulic and thermoelectric cooling techniques, which respectively adopt working fluids and electricity as coolants, are not economically feasible¹⁷. PCMs absorb heat by phase change characteristics at a certain temperature, which effectively cools the PV cells. It is expensive and not utilized industrially¹⁸.

After (Page 3-4):

Solar energy is widely utilized to reduce the grid's energy load¹. Photovoltaic (PV) cells convert energy from the sun directly to electricity, and are simple to set up because they have few components. The efficiency of a PV cell is approximately 10–25% considering that it can solely utilize a specific range of wavelengths. To date, although highly efficient PV cells have been developed including the perovskite cell², building-integrated PV modules are mostly composed of monocrystalline silicon with high technical maturity³. On the other hand, the power efficiency of PV devices is significantly affected by the cell temperature⁴. For instance, the power efficiency of monocrystalline PV is reduced by 33.3% when the junction temperature is increased from 20 °C to 60 °C⁵. A greater reduction in efficiency by the temperature increase has also been reported for the perovskite cells⁶. During electricity generation, the temperature rise owing to solar radiation is inevitable, and various cooling devices have been developed for stable power generation including passive cooling and active cooling strategies^{7,8}.

Passive cooling methods generate cooling effect permanently without energy consumption⁹: Natural air cooling is a phenomenon in which convective heat transfer occurs, owing to the density difference when the temperature of air rises around the cell. Elminshawy et al.¹⁰ reduced the average working temperature of PV cell by 19% utilizing the natural air cooling with attached fins at a submerged area ratio of 20%. Natural liquid cooling composed of PV modules and a fluid storage tank was studied¹¹. The cooling effect of natural liquid cooling can be improved by using nanofluid as a coolant in microchannels¹². The phase change cooling technologies have been actively studied utilizing two-phase coolant with boiling temperature of 34 °C¹³, RT35¹⁴, Al₂O₃ composites¹⁵, and paraffin-based materials¹⁶. The heat pipe cooling device has been proposed and a maximum thermal efficiency can be improved by 14.2%¹⁷. Li et al.¹⁸ developed atmospheric water sorption-based cooling for PV cell, which can also harvest water during power generation. The radiant cooling methods have been also discussed¹⁹. Li et al. studied the nighttime radiative cooling for PV cell with water harvesting²⁰. Recently, co-generation of electricity and freshwater²¹, dynamic photovoltaic building envelopes²², and PV cooling-driven seawater desalination²³ were discussed as the PV cooling strategies.

Active cooling methods consume energy to achieve a cooling effect: Forced air cooling generates air velocity through a fan, which has a better cooling effect than the natural convection²⁴. However, when the ambient temperature is high, the cooling effect significantly decreases, and the maintenance cost becomes high as well⁹. In addition, other challenges, such as space efficiency and energy consumption, make it less applicable. Hydraulic and thermoelectric cooling techniques, which respectively adopt working fluids and electricity as coolants, are not economically feasible²⁵. On the other hand, PV thermal systems (PVTs) have been developed. As the fluid circulates at the bottom of PV cell through the channels, the thermal energy is harvested and PV cell cools down. PVTs are feasible in that the harvested thermal energy can be utilized in buildings, although the energy is consumed to circulate fluid. The Al₂O₃ nanofluid²⁶, SiO₂ nanofluid²⁷, and microencapsulated PCM slurry fluid²⁸ were discussed as working fluids. Weinstein et al.²⁹ proposed beam splitting-based hybrid electric and thermal solar receiver.

8 Verma, S., Mohapatra, S., Chowdhury, S., & Dwivedi, G. Cooling techniques of the PV module: A review. *Materials Today: Proceedings* **38**, 253-258 (2021)

9 Ghadikolaei, S. S. C. An enviroeconomic review of the solar PV cells cooling technology effect on the CO₂ emission reduction. *Solar Energy*, **216**, 468-492 (2021)

2) The manuscript compares the performance of the presented MS-ER technique against air-cooled PV cells which is not a fair comparison. It is known that air-cooling of photovoltaic cells is the least efficient method. A good comparison would have been against phase change materials such as RT55, RT28HC, OM29, OM-47 etc.

(<https://doi.org/10.1016/j.matpr.2020.07.130>) MS-ER is basically a combination of moisture sorption/desorption (similar to phase change) and dissolution. Thus, comparison should be done against phase change materials.

Answer: Thank you for the good suggestion. The comparison of water desorption-driven endothermic reaction (WD-ER) cooling with other technologies was performed in Table 1. The cooling energy density of WD-ER reaches 2,876 kJ/kg (PCMs present 100-300 kJ/kg and moisture sorption-driven cooling presents 1,950 kJ/kg (Wang et al. Joule 2020)). The cooling power of WD-ER is 403 W/m², which outperforms radiative cooling methods of 40.1 W/m² (Raman et al. Nature 2020) and 310 W/m² (Wang et al. Joule 2020).

(Page 16)

Table 1 compares the cooling performance of WD-ER unit with other methods reported in the literature^{18,31,40-51}. The cooling energy density of PCMs ranges 100-300 kJ/kg. Although the temperature of PCMs is maintained at the melting point during the cooling process, the sensible heat change of the PV cell occurs continuously (i.e., the temperature of PV cell is not kept constant regardless of the state of PCMs). Moreover, the cooling energy density of MIL-101(Cr) coating layer is high at 1,950 kJ/kg using ambient moisture sorption-desorption process for electronic device applications³¹. Similarly, the PV cooling by atmospheric water sorption-evaporation cycle is proposed, and the cooling power is 295 W/m² with temperature drop of 10 °C. The radiative methods present cooling power of 40-300 W/m². The WD-ER of the present study outperforms the literature cooling methods in terms of cooling energy density (2,876 kJ/kg) and cooling power (403 W/m²).

Table 1. Comparison of WD-ER cooling unit with passive cooling methods

Methods		Density (kJ/kg)	Conditions and remarks	References
WD-ER	13X/H ₂ O/NH ₄ NO ₃	2,876	Operating range: 20-70 °C Cooling power: 403 W/m ² Temperature drop: 15.1 °C	
Phase change materials	RT55	170	Melting: 51-57 °C	40
	RT28HC	220	Melting: 28 °C	41
	C-PCM	199	Melting: 21.8 °C	42
	OM47	196	Melting: 48 °C	43
	MF-PW30	139.8	Melting: 56.8 °C crystallization: 45.1 °C	44
	PCMB	141.7	Steady range: 40.4-84.9 °C	45
	GO-EHS/PAAAM	200.3	Melting: 23 °C	46
	Eicosane	237.4	Melting: 36.5 °C	47
	Tricosane	269.2	Melting: 42-48 °C	48
	MIL-101(Cr)	1,950	Operating range: 20-70 °C	31
PAN-CNT-CaCl ₂	-	Operating range: 20-70 °C Cooling power: 295 W/m ² Temperature drop: 10.0 °C	18	
Radiative cooling	HfO ₂ and SiO ₂	-	Operating range: 22 °C Cooling power: 40.1 W/m ²	49
	Cooling assembly system	-	Operating range: 74 °C Cooling power: 310 W/m ² Temperature drop: 36.6 °C	50
	Silica pyramid	-	Operating range: 20-70 °C Temperature drop: 18.3 °C	51

- 40 Caliskan, H., Gurbuz, H., Sohret, Y. & Ates, D. Thermal analysis and assessment of phase change material utilization for heating applications in buildings: A modelling. *Journal of Energy Storage* **50**, 104593 (2022).
- 41 Zhao, Y. *et al.* Expanded graphite–Paraffin composite phase change materials: Effect of particle size on the composite structure and properties. *Applied Thermal Engineering* **171**, 115015 (2020).
- 42 Velmurugan, K. *et al.* Experimental studies on photovoltaic module temperature reduction using eutectic cold phase change material. *Solar Energy* **209**, 302-315 (2020).
- 43 Malvika, A., Arunachala, U. & Varun, K. Sustainable passive cooling strategy for photovoltaic module using burlap fabric-gravity assisted flow: A comparative Energy, exergy, economic, and enviroeconomic analysis. *Applied Energy* **326**, 120036 (2022).
- 44 Jing, J. H. *et al.* Melamine foam-supported form-stable phase change materials with simultaneous thermal energy storage and shape memory properties for thermal management of electronic devices. *ACS Applied Materials & Interfaces* **11**, 19252-19259 (2019).
- 45 Wu, W. *et al.* Preparation and thermal conductivity enhancement of composite phase change materials for electronic thermal management. *Energy Conversion and Management* **101**, 278-284 (2015).
- 46 Liu, Y., Yang, Y. & Li, S. Graphene oxide modified hydrate salt hydrogels: form-stable phase change materials for smart thermal management. *Journal of Materials Chemistry*

- A 4, 18134-18143 (2016).
- 47 Baby, R. & Balaji, C. Experimental investigations on phase change material based finned heat sinks for electronic equipment cooling. *International Journal of Heat and Mass Transfer* **55**, 1642-1649 (2012).
- 48 Weng, Y. C., Cho, H. P., Chang, C. C. & Chen, S. L. Heat pipe with PCM for electronic cooling. *Applied Energy* **88**, 1825-1833 (2011).
- 49 Raman, A. P., Anoma, M. A., Zhu, L., Rephaeli, E. & Fan, S. Passive radiative cooling below ambient air temperature under direct sunlight. *Nature* **515**, 540-544 (2014).
- 50 Wang, Z. *et al.* Lightweight, passive radiative cooling to enhance concentrating photovoltaics. *Joule* **4**, 2702-2717 (2020).
- 51 Zhu, L., Raman, A., Wang, K. X., Abou Anoma, M. & Fan, S. Radiative cooling of solar cells. *Optica* **1**, 32-38 (2014).

3) Compared to air-cooled photovoltaic cells only marginal improvements were achieved such as reduction in power generation cost by 5% over a period of 10 years and improvement in power efficiency by 7.5%. Thus, the manuscript needs to answer the big question “Why should MS-ER approach be employed in the first place when phase change materials result in better power efficiencies and benefits?” The easiest way to answer this critical question is to add a table of comparison in the manuscript which compares the MS-ER method against other PV cell cooling methods such as the use of phase change materials, use of cooling fluids and use of Peltier cells.

Answer: Thank you for your good comment. As mentioned above, instead of performing an economic analysis, Table 1 emphasizes the performance aspect of WD-ER unit by comparing it with the cooling methods reported in the literature. In addition, there are advantages such as increased space efficiency and durability of PV cells. If other high-efficiency PV devices are commercialized in the future, PV thermal management will become an even more important factor (perovskite cell has a greater efficiency decrease with temperature rise than that of monocrystal silicon).

Table 1 was added in the revised manuscript as follows;

Table 1. Comparison of WD-ER cooling unit with passive cooling methods

Methods		Density (kJ/kg)	Conditions and remarks	References
WD-ER	13X/H ₂ O/NH ₄ NO ₃	2,876	Operating range: 20-70 °C Cooling power: 403 W/m ² Temperature drop: 15.1 °C	
Phase change materials	RT55	170	Melting: 51-57 °C	40
	RT28HC	220	Melting: 28 °C	41
	C-PCM	199	Melting: 21.8 °C	42
	OM47	196	Melting: 48 °C	43
	MF-PW30	139.8	Melting: 56.8 °C crystallization: 45.1 °C	44
	PCMB	141.7	Steady range: 40.4-84.9 °C	45
	GO-EHS/PAAAM	200.3	Melting: 23 °C	46
Eicosane	237.4	Melting: 36.5 °C	47	

	Tricosane	269.2	Melting: 42-48 °C	48
	MIL-101(Cr)	1,950	Operating range: 20-70 °C	31
	PAN-CNT-CaCl ₂	-	Operating range: 20-70 °C Cooling power: 295 W/m ² Temperature drop: 10.0 °C	18
Radiative cooling	HfO ₂ and SiO ₂	-	Operating range: 22 °C Cooling power: 40.1 W/m ²	49
	Cooling assembly system	-	Operating range: 74 °C Cooling power: 310 W/m ² Temperature drop: 36.6 °C	50
	Silica pyramid	-	Operating range: 20-70 °C Temperature drop: 18.3 °C	51

4) The manuscript claims self-healing passive cooling using endothermic reaction. However, the self-healing aspect was not demonstrated. It is ideal that the authors present the performance of the fabricated PV cell after repeated cycles of ammonium nitrate dissolution and recrystallization. The self-healing is easier claimed than demonstrated. Recrystallization of ammonium nitrate crystals after temperature reduction during night-time is not guaranteed to occur along a thin film format. Natural crystallization can occur in the form of elongated and large crystals which will reduce the efficiency of dissolution after each cycle. The cycling performance is an essential requirement to satisfy the claim of self-healing.

Answer: The term “self-healing” was replaced by “self-recovering” in the revised manuscript. The self-recovering aspect was discussed in lab-scale heat dissipation performance, based on the film composed of water-saturated zeolite 13X and ammonium nitrate layer. The natural crystallization phenomenon was also considered based on the cyclic data of Figure 5(f) as follow;

Before:

It is evaluated that after the MS-ER cooling unit performs heat dissipation, whether all processes are self-healed (reversible process) by cyclic experiments when the temperature is lowered. In Figure 7f, as the effective heat transfer coefficient is maintained at 64.1–68.9 W/m²·K in five cycles, it is concluded that the MS-ER cooling unit exhibits self-healing characteristics. Specifically, at 20 °C, zeolite 13X completely adsorbs water again, NH₄NO₃ crystallizes to form a hard layer, and MS-ER exhibits a solid-like structure. When heat is supplied, water is regenerated from zeolite 13X, and the water dissolves NH₄NO₃ again to generate a cooling effect. Accordingly, at 50 °C, an aqueous solution in which NH₄⁺ and NO₃⁻ are dissolved is observed, and the color of zeolite 13X is pale, indicating that water is desorbed.

After (Page 13-14):

To further discuss the heat dissipation mechanism, the average effective heat transfer coefficient of the WD-ER unit (13X/H₂O/NH₄NO₃) is 64.1 W/m²·K, far superior to the sum of effective heat transfer coefficients of latent heat cooling utilizing 13X/H₂O (29 W/m²·K) and endothermic cooling utilizing NH₄NO₃/H₂O (19.5 W/m²·K). The latent cooling mechanism of WD-ER is invariant compared to that of 13X/H₂O in that the thermal energy supplied to the PV is transferred to the thin film and it induces water desorption. However, the WD-ER

presents a different aspect in terms of dissolution process when only endothermic reaction cooling ($\text{NH}_4\text{NO}_3/\text{H}_2\text{O}$) is performed. As thermal energy is directly supplied to $\text{NH}_4\text{NO}_3/\text{H}_2\text{O}$, the solubility increases and NH_4NO_3 is dissolved, resulting in an endothermic reaction. In this process, the reaction area between NH_4NO_3 and H_2O is limited in that water cannot flow between the NH_4NO_3 crystal layers. When endothermic cooling is induced after the latent cooling for WD-ER, water molecules generated in the form of droplets permeate between the NH_4NO_3 crystal layers and the reaction area widens (in Figure S3, the water droplets are generated on the surface of zeolite 13X). In summary, the cooling power of WD-ER increases due to the improvement of dissolution process by widening the interfacial area. On the other hand, the cooling energy density of WD-ER (2,876 kJ/kg- H_2O) is similar to the sum of cooling energy densities of latent cooling (346.6 kJ/kg-13X is equivalent to 1,815 kJ/kg- H_2O) and endothermic reaction cooling (1,386 kJ/kg- H_2O) with a 10% error.

It is evaluated that after the **WD-ER** cooling unit performs heat dissipation, whether all processes are self-recovered by cyclic experiments. In Figure 5(f), as the effective heat transfer coefficient is maintained at 64.1–68.9 $\text{W}/\text{m}^2\cdot\text{K}$ in five cycles, it is concluded that the WD-ER cooling unit exhibits self-recovering characteristics. Specifically, at 20 °C, zeolite 13X completely adsorbs water again, NH_4NO_3 crystallizes to form a hard layer, and **WD-ER** exhibits a solid-like structure. When heat is supplied, water is regenerated from zeolite 13X, and the water dissolves NH_4NO_3 again to generate a cooling effect. Accordingly, at 50 °C, an aqueous solution in which NH_4^+ and NO_3^- are dissolved is observed, and the color of zeolite 13X is pale, indicating that water is desorbed. **Note that, in the solidification process of NH_4NO_3 , crystals could be randomly formed. However, the reproducibility problem due to the random crystallization does not occur in Figure 5(f) because the interfacial area of dissolution is large as the water droplets desorbed from zeolite 13X permeate between them.**

Figure 5. Heat dissipation performance in terms of cooling methods; (f) Repeated WD-ER cooling.

In addition, the average cooling power density and temperature variation during the day were evaluated in Figure 8(a), when the outdoor temperature is 30.5-31.7 °C (Page 15-16).

The cyclic cooling performance of WD-ER is evaluated in Figure 8(a), when the outdoor temperature is 30.5-31.7 °C. During the power generation period, the temperature of PV cell with WD-ER reaches a maximum of 60 °C, and after the sunset, the temperature drops to around 25 °C, confirming the self-recovering characteristics (a temperature difference of 35 °C, which is maintained for more than 8 hours, is sufficient for recovering process. In addition,

the radiation cooling effect is also significant at night.). The average cooling power density is uniform at 375-419 W/m², although it slightly differs in terms of external conditions. In winter, the cooling effect of the WD-ER unit could be more important as the working temperature range becomes larger (In general, the temperature range of PV cell is 0-55 °C during power generation, and at night, the temperature drops to -10 °C). Figure 8(b) illustrates the current-voltage characteristics of the monocrystalline silicon. As the temperature rises, the open-circuit voltage increases, while the short-circuit current decreases. The maximum power is inversely proportional to temperature (25 °C: 91.3 mW → 60 °C: 81.4 mW). In summary, the power efficiency of PV cell with WD-ER unit is improved by approximately 10% compared to that of PV cell with natural air cooling. The improvement of power efficiency has been analyzed based on a commercially available monocrystalline silicon.

Figure 8. Practical working performance of WD-ER unit; (a) Cooling power density and average temperature variation during the day; (b) Current-voltage characteristics in terms of temperature.

Reviewer #3 (Remarks to the Author):

Title: Self-healing passive cooling utilizing endothermic reaction of NH₄NO₃/H₂O driven by moisture sorption for photovoltaic cell

Comments:

A cooling method that combines latent cooling and endothermic reaction cooling is proposed to cool solar cells for PV applications. According to the manuscript, the average temperature of the PV cell can be reduced by 14.9 °C, corresponding to an efficiency improvement of 7.5%. In addition, the cost is also reduced by 5.1%. The new contribution and actual cooling effect of the proposed method should be re-checked. Furthermore, I do not think the impact warrants publication in Nature Communications. Please find my comments below.

Answer: Thank you for your valuable comments and efforts in evaluating the manuscript. The manuscript was faithfully modified to reflect all comments.

1. While the combined cooling unit for PV cooling is a new structure, I question the performance and true impact of the proposed cooling method. The proposed cooling method may be general in the field of thermodynamics, what's the main highlight of this method when compared to other similar cooling methods, such as PCM cooling.

Answer: Thanks for the advice. The novelty of water desorption-driven endothermic reaction cooling (WD-ER) technology was emphasized by comparing it to the latest reported cooling method including moisture-sorption cooling (Wang et al. Joule 2020) and water evaporation cooling (Le et al. Nature sustainability 2020). It was incorporated in the revised manuscript as follows;

Before:

In summary, while low-cost cooling methods exhibit a low cooling effect, the methods with high cooling efficiencies are expensive. Herein, a novel type of passive cooling unit for a PV cell is proposed, and presents the best performance among cooling methods that have been reported in the literature^{19, 20}. The method has no energy consumption and consists of cheap materials, making it highly applicable. The temperature of the cell decreases significantly during power generation by applying a passive cooling unit, and it can be repeatedly utilized, owing to its self-heating characteristics. The cooling mechanisms of the passive cooling unit have been investigated, and material design is possible for optimal cooling energy capacity, considering the operational conditions. Based on the feasibility study, it is economically feasible for industrial-scale applications with a highly efficient cooling performance.

In INTRODUCTION (Page 4)

To summarize, while low-cost cooling methods exhibit a low cooling effect, the methods with high cooling efficiencies are expensive at the present level of technologies. Here, a novel type of passive cooling unit for a PV cell is proposed to realize the best performance among the cooling methods that have been reported in the literature^{9,30}. The cooling system induced by a chain reaction of latent cooling (water desorption process) and endothermic reaction (dissolution process) has not been attempted for PV applications, which is the main novelty of this study. The proposed method has no energy consumption and consists of cheap materials, making it highly applicable. Moreover, it can be repeatedly utilized, owing to its self-recovering characteristics. Meanwhile, a similar thermal management strategy for electronic devices was reported³¹. As heat is generated in electronic devices, the water molecules trapped in metal-organic frameworks (MOFs) are desorbed, which produces latent cooling effect. In this study, we have improved the atmospheric water desorption-driven cooling system by the chain reaction in which the desorbed water melts the crystal layer of ammonium nitrate and the secondary endothermic reaction cooling in a closed thin film.

The physical working principle of WD-ER unit was discussed additionally to focus on main contributions of present method.

The deeper reflection on the physical working principle of WD-ER unit is described. The water desorption process from zeolite 13X is expressed as Eq. (1), where Q_d is desorption

energy (kJ), m_a is adsorbent mass (kg), h_d is average desorption enthalpy of water (kJ/kg-H₂O), and x_w is adsorption capacity (kg-H₂O/kg-13X). The subscripts i, and f represent the initial and final states, respectively.

$$Q_d = m_a \cdot h_d \cdot (x_{w,i} - x_{w,f}) \quad (1)$$

In the heat exchange process between the PV cell and the thin film (WD-ER unit), the Biot number is less than 0.01, which supports the lumped system analysis in the longitudinal direction, and most of the radiant energy is consumed for water desorption. Moreover, the solar thermal energy is uniformly supplied to the PV modules, enabling one-dimensional analysis. The well-known diffusion equation of desorbed water is Eq. (2)³¹, where m_w is desorbed water mass (kg), K^* is modified mass transfer coefficient (m/s), A is cross-sectional area (m²), C_s is concentration of water on the particle surface (kg/m³), and C_f is free stream concentration (kg/m³), respectively (This equation is defined as a pseudo first-order kinetic model³⁵).

$$\frac{d(m_w)}{dt} = K^* \cdot A \cdot (C_s - C_f) \quad (2)$$

However, in the case of WD-ER unit, the diffusion of water molecules is limited in that the concentration difference is not significant. Although the desorption enthalpy of water from microporous materials is lower than that of water vaporization, the cooling effect is estimated as high as 27.4 kJ/mol-H₂O.

The endothermic reaction cooling of dissolution process is estimated by Eq. (3), where Q_e is endothermic reaction energy (kJ) and x_s is adsorption capacity (kg-NH₄NO₃/kg-H₂O). The subscript s represent solute.

$$Q_e = m_w \cdot h_e \cdot (x_{s,f} - x_{s,i}) \quad (3)$$

The dissolution enthalpy (h_e ; kJ/kg-solute) is the difference between the formation enthalpies of the aqueous and solid phases. The ideal solubility is defined as the saturated mole fraction of the solute in the solution. It is evaluated by the equilibrium of the chemical potential between the solid and solution states as Eq. (4), where y is the mole fraction of solute in the solution, R is the gas constant, T is temperature, and T_F is the freezing temperature of the solute, respectively.

$$\ln(y) = \frac{h_e}{R} \left(\frac{1}{T} - \frac{1}{T_F} \right) \quad (4)$$

The total cooling energy density of WD-ER unit can be obtained by considering the chain reaction of water desorption and solute dissolution.

The discussion on cooling performance of WD-ER unit was supplemented (Page 13-14)

To further discuss the heat dissipation mechanism, the average effective heat transfer coefficient of the WD-ER unit (13X/H₂O/NH₄NO₃) is 64.1 W/m²·K, far superior to the sum of effective heat transfer coefficients of latent heat cooling utilizing 13X/H₂O (29 W/m²·K) and endothermic cooling utilizing NH₄NO₃/H₂O (19.5 W/m²·K). The latent cooling mechanism of WD-ER is invariant compared to that of 13X/H₂O in that the thermal energy supplied to the PV is transferred to the thin film and it induces water desorption. However, the WD-ER presents a different aspect in terms of dissolution process when only endothermic reaction cooling (NH₄NO₃/H₂O) is performed. As thermal energy is directly supplied to NH₄NO₃/H₂O,

the solubility increases and NH_4NO_3 is dissolved, resulting in an endothermic reaction. In this process, the reaction area between NH_4NO_3 and H_2O is limited in that water cannot flow between the NH_4NO_3 crystal layers. When endothermic cooling is induced after the latent cooling for WD-ER, water molecules generated in the form of droplets permeate between the NH_4NO_3 crystal layers and the reaction area widens (in Figure S3, the water droplets are generated on the surface of zeolite 13X). In summary, the cooling power of WD-ER increases due to the improvement of dissolution process by widening the interfacial area. On the other hand, the cooling energy density of WD-ER (2,876 kJ/kg- H_2O) is similar to the sum of cooling energy densities of latent cooling (346.6 kJ/kg-13X is equivalent to 1,815 kJ/kg- H_2O) and endothermic reaction cooling (1,386 kJ/kg- H_2O) with a 10% error.

More importantly, the cooling performance of WD-ER unit was compared with other technologies in Table 1.

Table 1 compares the cooling performance of WD-ER unit with other methods reported in the literature^{18,31,40-51}. The cooling energy density of PCMs ranges 100-300 kJ/kg. Although the temperature of PCMs is maintained at the melting point during the cooling process, the sensible heat change of the PV cell occurs continuously (i.e., the temperature of PV cell is not kept constant regardless of the state of PCMs). Moreover, the cooling energy density of MIL-101(Cr) coating layer is high at 1,950 kJ/kg using ambient moisture sorption-desorption process for electronic device applications³¹. Similarly, the PV cooling by atmospheric water sorption-evaporation cycle is proposed, and the cooling power is 295 W/m² with temperature drop of 10 °C. The radiative methods present cooling power of 40-300 W/m². The WD-ER of the present study outperforms the literature cooling methods in terms of cooling energy density (2,876 kJ/kg) and cooling power (403 W/m²).

Table 1. Comparison of WD-ER cooling unit with passive cooling methods

Methods		Density (kJ/kg)	Conditions and remarks	References
WD-ER	13X/ H_2O / NH_4NO_3	2,876	Operating range: 20-70 °C Cooling power: 403 W/m ² Temperature drop: 15.1 °C	
Phase change materials	RT55	170	Melting: 51-57 °C	40
	RT28HC	220	Melting: 28 °C	41
	C-PCM	199	Melting: 21.8 °C	42
	OM47	196	Melting: 48 °C	43
	MF-PW30	139.8	Melting: 56.8 °C crystallization: 45.1 °C	44
	PCMB	141.7	Steady range: 40.4-84.9 °C	45
	GO-EHS/PAAAM	200.3	Melting: 23 °C	46
	Eicosane	237.4	Melting: 36.5 °C	47
	Tricosane	269.2	Melting: 42-48 °C	48
	MIL-101(Cr)	1,950	Operating range: 20-70 °C	31
	PAN-CNT- CaCl_2	-	Operating range: 20-70 °C Cooling power: 295 W/m ² Temperature drop: 10.0 °C	18
Radiative	HfO_2 and SiO_2	-	Operating range: 22 °C	49

cooling			Cooling power: 40.1 W/m ²	
	Cooling assembly system	-	Operating range: 74 °C Cooling power: 310 W/m ² Temperature drop: 36.6 °C	50
	Silica pyramid	-	Operating range: 20-70 °C Temperature drop: 18.3 °C	51

- 40 Caliskan, H., Gurbuz, H., Sohret, Y. & Ates, D. Thermal analysis and assessment of phase change material utilization for heating applications in buildings: A modelling. *Journal of Energy Storage* **50**, 104593 (2022).
- 41 Zhao, Y. *et al.* Expanded graphite–Paraffin composite phase change materials: Effect of particle size on the composite structure and properties. *Applied Thermal Engineering* **171**, 115015 (2020).
- 42 Velmurugan, K. *et al.* Experimental studies on photovoltaic module temperature reduction using eutectic cold phase change material. *Solar Energy* **209**, 302-315 (2020).
- 43 Malvika, A., Arunachala, U. & Varun, K. Sustainable passive cooling strategy for photovoltaic module using burlap fabric-gravity assisted flow: A comparative Energy, exergy, economic, and enviroeconomic analysis. *Applied Energy* **326**, 120036 (2022).
- 44 Jing, J. H. *et al.* Melamine foam-supported form-stable phase change materials with simultaneous thermal energy storage and shape memory properties for thermal management of electronic devices. *ACS Applied Materials & Interfaces* **11**, 19252-19259 (2019).
- 45 Wu, W. *et al.* Preparation and thermal conductivity enhancement of composite phase change materials for electronic thermal management. *Energy Conversion and Management* **101**, 278-284 (2015).
- 46 Liu, Y., Yang, Y. & Li, S. Graphene oxide modified hydrate salt hydrogels: form-stable phase change materials for smart thermal management. *Journal of Materials Chemistry A* **4**, 18134-18143 (2016).
- 47 Baby, R. & Balaji, C. Experimental investigations on phase change material based finned heat sinks for electronic equipment cooling. *International Journal of Heat and Mass Transfer* **55**, 1642-1649 (2012).
- 48 Weng, Y. C., Cho, H. P., Chang, C. C. & Chen, S. L. Heat pipe with PCM for electronic cooling. *Applied Energy* **88**, 1825-1833 (2011).
- 49 Raman, A. P., Anoma, M. A., Zhu, L., Rephaeli, E. & Fan, S. Passive radiative cooling below ambient air temperature under direct sunlight. *Nature* **515**, 540-544 (2014).
- 50 Wang, Z. *et al.* Lightweight, passive radiative cooling to enhance concentrating photovoltaics. *Joule* **4**, 2702-2717 (2020).
- 51 Zhu, L., Raman, A., Wang, K. X., Abou Anoma, M. & Fan, S. Radiative cooling of solar cells. *Optica* **1**, 32-38 (2014).

2.The experimental testing performed in the study is just a case study. Adsorption-desorption process and dissolution-crystallization process need different driven parameters (e.g., ambient temperature), so please explore the cooling performance under different weather conditions, such as in the winter season and summer seasons. Thus, the applicability of the combined cooling method can be proved.

Answer: Thank for your comments. The adsorption-desorption cycles (including dissolution-crystallization process) were performed as presented in Figure 8(a).

The cyclic cooling performance of WD-ER is evaluated in Figure 8(a), when the

outdoor temperature is 30.5-31.7 °C. During the power generation period, the temperature of PV cell with WD-ER reaches a maximum of 60 °C, and after the sunset, the temperature drops to around 25 °C, confirming the self-recovering characteristics (a temperature difference of 35 °C, which is maintained for more than 8 hours, is sufficient for recovering process. In addition, the radiation cooling effect is also significant at night.). The average cooling power density is uniform at 375-419 W/m², although it slightly differs in terms of external conditions. In winter, the cooling effect of the WD-ER unit could be more important as the working temperature range becomes larger (In general, the temperature range of PV cell is 0-55 °C during power generation, and at night, the temperature drops to -10 °C). Figure 8(b) illustrates the current-voltage characteristics of the monocrystalline silicon. As the temperature rises, the open-circuit voltage increases, while the short-circuit current decreases. The maximum power is inversely proportional to temperature (25 °C: 91.3 mW → 60 °C: 81.4 mW). In summary, the power efficiency of PV cell with WD-ER unit is improved by approximately 10% compared to that of PV cell with natural air cooling. The improvement of power efficiency has been analyzed based on a commercially available monocrystalline silicon.

Figure 8. Practical working performance of WD-ER unit; (a) Cooling power density and average temperature variation during the day; (b) Current-voltage characteristics in terms of temperature.

To summarize, it presents steady performance based on the cooling power density and average temperature difference under extreme operating conditions (summer with the high humidity). The self-recovering characteristics have been verified, and should work in other environments, because it is driven by internal chain reaction regardless of the type of heat supplied and ambient conditions.

As for the cooling performance under different weather conditions, such as in the winter season and summer seasons, the cooling performance will be more significantly improved in winter because the working temperature of the PV cell ranges about 5-60 °C in winter, leading to a larger temperature difference than that in summer (about 30-70 °C).

3.The detailed analysis for cost is missing. It is very important to conduct a detailed cost analysis since the cost of PV applications is decreasing with a low price. So, the cost may determine the reasonability of the cooling method for industrial applications.

Answer: Thanks for the good point. The main objective of this study is to improve the cooling performance of the WD-ER by the chain reaction of latent cooling (water desorption process) and endothermic reaction (dissolution process) for PV applications, which is the main novelty of this study. Therefore, instead of performing a detailed cost analysis of photovoltaic cells with WD-ER unit, the manuscript was revised to concentrate on the cooling performance analysis (In Table 1, the cooling performance of the WD-ER unit is far superior to that reported in the literature). Based on this, the economic feasibility will be much better when it is applied to high-efficiency cells including perovskite materials in the future. In addition, zeolite 13X, ammonium nitrate and water are much cheaper than PCMs.

It was incorporated in the revised manuscript as follows on p. 4.

To summarize, while low-cost cooling methods exhibit a low cooling effect, the methods with high cooling efficiencies are expensive at the present level of technologies. Here, a novel type of passive cooling unit for a PV cell is proposed to realize the best performance among the cooling methods that have been reported in the literature^{9,30}. The cooling system induced by a chain reaction of latent cooling (water desorption process) and endothermic reaction (dissolution process) has not been attempted for PV applications, which is the main novelty of this study. The proposed method has no energy consumption and consists of cheap materials, making it highly applicable. Moreover, it can be repeatedly utilized, owing to its self-recovering characteristics. Meanwhile, a similar thermal management strategy for electronic devices was reported³¹. As heat is generated in electronic devices, the water molecules trapped in metal-organic frameworks (MOFs) are desorbed, which produces latent cooling effect. In this study, we have improved the atmospheric water desorption-driven cooling system by the chain reaction in which the desorbed water melts the crystal layer of ammonium nitrate and the secondary endothermic reaction cooling in a closed thin film.

4. For equation (5), does h_{eff} consider the effect of radiation heat transfer? In addition, during the heat dissipation experiment, the apparatus should be placed outside so that radiation heat transfer between the cell and the cold sky occurs. Notably, radiative heat dissipation of the cell is also an important cooling method, which will be suppressed in the indoor environment.

Answer: The lab-scale heat dissipation experiments were conducted indoors to create precise experimental conditions, and therefore the radiation heat transfer was negligible in equation (5). However, the radiation effect was included for the outdoor experiments in Fig. 7 and Fig. 8. It was incorporated in the revised manuscript as follows;

The heat dissipation performance is evaluated with Figure S4. The simple energy balance is defined in Eq. (5) by analyzing the cell as the control volume to compare the cooling methods. m is the mass of the cell (kg), C_p is the specific heat capacity of the cell (kJ/kg·K), \dot{Q} is the supplied heat transfer rate (kW), Δt is the time interval between steps (s), h_{eff} is the effective heat transfer coefficient (W/m²·K), A is the cross-sectional area (m²), $T_{s,i}$ is the cell temperature at the i^{th} interval (°C), and T_a is the ambient temperature (°C).

$$m \cdot C_p \cdot (T_{s,i+1} - T_{s,i}) = \dot{Q} \cdot \Delta t - h_{eff} \cdot A \cdot (T_{s,i} - T_a) \cdot \Delta t \quad (5)$$

Although the WD-ER cooling unit extracts thermal energy by latent heat of porous material and endothermic reaction of solute layer, a convective heat transfer model utilizing effective heat transfer coefficient of Eq. (5) is applied for quantitative comparison. Meanwhile, the radiant heat transfer can be neglected in that the heat dissipation experiment was conducted

indoors. However, the operation of WD-ER cooling unit is invariant regardless of the form of external heat transfer. The Biot number of natural air cooling is $4 \cdot 10^{-5}$ which makes it possible to neglect the spatial temperature distribution.

More importantly, the real applications of WD-ER unit was considered as presented in Figure 7 (transient performance during the day) and Figure 8 (cycle performances), which includes the radiation heat transfer effect on page 14.

The WD-ER cooling unit is applied to a PV cell comprising monocrystalline silicon. As illustrated in Figure 7(a), urethane waterproofing on the back of the PV cell protects the power terminal unit from moisture penetration. Zeolite 13X particles are coated on the PV cell with a thickness of 2 mm, and a negligible amount of polyvinyl acetate is included. After the coated zeolite 13X is saturated with water, the NH_4NO_3 crystal layer is added to configure the WD-ER cooling unit (The configuration of PV cell with WD-ER cooling unit for industrial applications is same as that of lab-scale heat dissipation experiment in Figure 5). PV cells with natural air cooling and WD-ER cooling units were installed in adjacent positions to observe the temperature variation. Based on the images taken with a thermal imaging camera at 2 pm, the surface temperature of the PV cell with the WD-ER cooling unit was 39–43°C, which is significantly lower than that of natural air cooling (51–56 °C) as shown in Figure 7(a). Figure 7(b) presents the temperature variation during the power generation process of the PV cell. The severe temperature fluctuation results from the phenomena such as intermittent winds inducing local forced air cooling and obscuring of sunlight by clouds. The temperature of the PV cell with WD-ER is much lower than that of natural air cooling, and the average temperature is reduced by 14.9 °C (from 59.1 °C to 44.2 °C).

Figure 7(c) illustrates the net cooling power density considering the temperature difference between natural air and WD-ER cooling (temperature data were converted into cooling energy per unit area of PV cell utilizing the smoothing filter of Savitzky-Golay theory. After dividing it by the measurement time interval, the cooling power density was obtained). When the cell temperature is in the range of 20–45 °C (time range of 0–3,500 seconds), the radiation heat is absorbed at 380 W/m², which corresponds to the desorption energy of water from zeolite 13X (latent cooling). The absorption heat amount rapidly increases to 495 W/m² at 45 °C as the NH_4NO_3 layer reacts with solvent, which is the endothermic reaction cooling for 3,500–4,000 seconds. It is similar to the case when WD-ER cooling unit is applied for lab-scale heat dissipation experiment in Figure 5(e), the temperature rapidly decreases around 45 °C. Further, the cooling power density is maintained at 431 W/m² with the combined water desorption and solute dissolution. When the cooling energy density reaches 2,500 kJ/kg at approximately 9,000 seconds, the cooling performance is significantly reduced as the WD-ER cooling unit begins to be saturated. Finally, the cooling energy density is 2,876 kJ/kg, which is sufficient for utilization under high ambient temperature conditions.

The cyclic cooling performance of WD-ER is evaluated in Figure 8(a), when the outdoor temperature is 30.5–31.7 °C. During the power generation period, the temperature of PV cell with WD-ER reaches a maximum of 60 °C, and after the sunset, the temperature drops to around 25 °C, confirming the self-recovering characteristics (a temperature difference of 35 °C, which is maintained for more than 8 hours, is sufficient for recovering process. In addition, the radiation cooling effect is also significant at night.). The average cooling power density is uniform at 375–419 W/m², although it slightly differs in terms of external conditions. In winter, the cooling effect of the WD-ER unit could be more important as the working temperature

range becomes larger (In general, the temperature range of PV cell is 0-55 °C during power generation, and at night, the temperature drops to -10 °C). Figure 8(b) illustrates the current-voltage characteristics of the monocrystalline silicon. As the temperature rises, the open-circuit voltage increases, while the short-circuit current decreases. The maximum power is inversely proportional to temperature (25 °C: 91.3 mW → 60 °C: 81.4 mW). In summary, the power efficiency of PV cell with WD-ER unit is improved by approximately 10% compared to that of PV cell with natural air cooling. The improvement of power efficiency has been analyzed based on a commercially available monocrystalline silicon.

Figure 7. Application of WD-ER unit for PV power generation; (b) Temperature variation of PV cell during a day; (c) Net cooling power density of WD-ER unit without natural convection.

Figure 8. Practical working performance of WD-ER unit; (a) Cooling power density and average temperature variation during the day; (b) Current-voltage characteristics in terms of temperature.

5. Please explore the thickness of the zeolite layer on the cooling performance since a thicker zeolite layer provides more latent cooling ability but introduces additional heat resistance.

Answer: The thickness variation of zeolite 13X layer was explored on the cooling performance in Figure 6. The heat dissipation kinetics are almost constant regardless of the thickness, but the cooling energy density is proportional to the thickness. Moreover, Biot number of WD-ER unit is in the range of $1.5-7.5 \times 10^{-5}$, and the thermal resistance can be neglected.

It is evaluated that after the **WD-ER** cooling unit performs heat dissipation, whether all processes are self-recovered by cyclic experiments. In Figure 5(f), as the effective heat transfer coefficient is maintained at 64.1–68.9 W/m²·K in five cycles, it is concluded that the WD-ER cooling unit exhibits self-recovering characteristics. Specifically, at 20 °C, zeolite 13X completely adsorbs water again, NH₄NO₃ crystallizes to form a hard layer, and **WD-ER** exhibits a solid-like structure. When heat is supplied, water is regenerated from zeolite 13X, and the water dissolves NH₄NO₃ again to generate a cooling effect. Accordingly, at 50 °C, an aqueous solution in which NH₄⁺ and NO₃⁻ are dissolved is observed, and the color of zeolite 13X is pale, indicating that water is desorbed. **Note that, in the solidification process of NH₄NO₃, crystals could be randomly formed. However, the reproducibility problem due to the random crystallization does not occur in Figure 5(f) because the interfacial area of dissolution is large as the water droplets desorbed from zeolite 13X permeate between them. Moreover, the thickness change of zeolite 13X film layer does not affect the heat dissipation kinetics (rate/behavior), and only the cooling energy density is varied as presented in Figure 6. As the temperature change is the driving force of water desorption, a certain amount of water molecules are regenerated regardless of the thickness of the layer (The rate-limiting step of water desorption process is dominated by the cross-sectional area instead of the thickness)³¹. As the thickness of zeolite 13X film is in the range of 1-5 mm, the thermal resistance can be neglected with the Biot number of 1.5-7.5 10⁻⁵.**

Figure 6. Heat dissipation performance of latent cooling in terms of zeolite film thickness

6. In Fig. 7, it seems that the testing configuration for natural and forced air cooling is quite different from that for latent cooling and endothermic reaction cooling, so the comparison of effective heat transfer coefficients is unfair. Besides, the temperature in Fig. 7c and d represents which components, cells, or other objects?

Answer: Thank you for your advice. The testing configurations for natural/forced air cooling, latent cooling, and endothermic reaction cooling were the same as presented in Figure S4. It seems that the reviewer misunderstood the visualization data (Figure S3) inserted into the Figure 6. The temperature data in Figure 6 represents temperature of the cell. The temperature in Fig. 5c (7c in old version) is for the cells.

7.A detailed Reversible cycle testing (e.g., long time, more cycles, more operation conditions) can be added to highlight the feature of the proposed strategy.

Answer: The lab-scale reversible cycle testing was discussed in Figure 5(f) as follows;

Before:

It is evaluated that after the MS-ER cooling unit performs heat dissipation, whether all processes are self-healed (reversible process) by cyclic experiments when the temperature is lowered. In Figure 7f, as the effective heat transfer coefficient is maintained at 64.1–68.9 W/m²·K in five cycles, it is concluded that the MS-ER cooling unit exhibits self-healing characteristics. Specifically, at 20 °C, zeolite 13X completely adsorbs water again, NH₄NO₃ crystallizes to form a hard layer, and MS-ER exhibits a solid-like structure. When heat is supplied, water is regenerated from zeolite 13X, and the water dissolves NH₄NO₃ again to generate a cooling effect. Accordingly, at 50 °C, an aqueous solution in which NH₄⁺ and NO₃⁻ are dissolved is observed, and the color of zeolite 13X is pale, indicating that water is desorbed.

After (Page 13-14):

To further discuss the heat dissipation mechanism, the average effective heat transfer coefficient of the WD-ER unit (13X/H₂O/NH₄NO₃) is 64.1 W/m²·K, far superior to the sum of effective heat transfer coefficients of latent heat cooling utilizing 13X/H₂O (29 W/m²·K) and endothermic cooling utilizing NH₄NO₃/H₂O (19.5 W/m²·K). The latent cooling mechanism of WD-ER is invariant compared to that of 13X/H₂O in that the thermal energy supplied to the PV is transferred to the thin film and it induces water desorption. However, the WD-ER presents a different aspect in terms of dissolution process when only endothermic reaction cooling (NH₄NO₃/H₂O) is performed. As thermal energy is directly supplied to NH₄NO₃/H₂O, the solubility increases and NH₄NO₃ is dissolved, resulting in an endothermic reaction. In this process, the reaction area between NH₄NO₃ and H₂O is limited in that water cannot flow between the NH₄NO₃ crystal layers. When endothermic cooling is induced after the latent cooling for WD-ER, water molecules generated in the form of droplets permeate between the NH₄NO₃ crystal layers and the reaction area widens (in Figure S3, the water droplets are generated on the surface of zeolite 13X). In summary, the cooling power of WD-ER increases due to the improvement of dissolution process by widening the interfacial area. On the other hand, the cooling energy density of WD-ER (2,876 kJ/kg-H₂O) is similar to the sum of cooling energy densities of latent cooling (346.6 kJ/kg-13X is equivalent to 1,815 kJ/kg-H₂O) and endothermic reaction cooling (1,386 kJ/kg-H₂O) with a 10% error.

It is evaluated that after the WD-ER cooling unit performs heat dissipation, whether all processes are self-recovered by cyclic experiments. In Figure 5(f), as the effective heat transfer coefficient is maintained at 64.1–68.9 W/m²·K in five cycles, it is concluded that the WD-ER cooling unit exhibits self-recovering characteristics. Specifically, at 20 °C, zeolite 13X completely adsorbs water again, NH₄NO₃ crystallizes to form a hard layer, and WD-ER exhibits a solid-like structure. When heat is supplied, water is regenerated from zeolite 13X, and the water dissolves NH₄NO₃ again to generate a cooling effect. Accordingly, at 50 °C, an aqueous solution in which NH₄⁺ and NO₃⁻ are dissolved is observed, and the color of zeolite 13X is pale, indicating that water is desorbed. Note that, in the solidification process of NH₄NO₃, crystals could be randomly formed. However, the reproducibility problem due to the random crystallization does not occur in Figure 5(f) because the interfacial area of dissolution is large as the water droplets desorbed from zeolite 13X permeate between them.

Figure 5. Heat dissipation performance in terms of cooling methods; (f) Repeated WD-ER cooling.

More importantly, the reversible cycle testing for practical applications was discussed in Figure 8.

The cyclic cooling performance of WD-ER is evaluated in Figure 8(a), when the outdoor temperature is 30.5-31.7 °C. During the power generation period, the temperature of PV cell with WD-ER reaches a maximum of 60 °C, and after the sunset, the temperature drops to around 25 °C, confirming the self-recovering characteristics (a temperature difference of 35 °C, which is maintained for more than 8 hours, is sufficient for recovering process. In addition, the radiation cooling effect is also significant at night.). The average cooling power density is uniform at 375-419 W/m², although it slightly differs in terms of external conditions. In winter, the cooling effect of the WD-ER unit could be more important as the working temperature range becomes larger (In general, the temperature range of PV cell is 0-55 °C during power generation, and at night, the temperature drops to -10 °C). Figure 8(b) illustrates the current-voltage characteristics of the monocrystalline silicon. As the temperature rises, the open-circuit voltage increases, while the short-circuit current decreases. The maximum power is inversely proportional to temperature (25 °C: 91.3 mW → 60 °C: 81.4 mW). In summary, the power efficiency of PV cell with WD-ER unit is improved by approximately 10% compared to that of PV cell with natural air cooling. The improvement of power efficiency has been analyzed based on a commercially available monocrystalline silicon.

Figure 8. Practical working performance of WD-ER unit; (a) Cooling power density and average temperature variation during the day; (b) Current-voltage characteristics in terms of temperature.

8. For outdoor testing (i.e., Fig. 8), the testing details should be given. For example, how to deal with the generated electricity? Generally, a control with a maximal power point tracking function should be used to collect the transient generated power. If generated power is dissipated into the heat of the cell, the temperature of the cell will rise, which affects the accuracy of the monitored data and related conclusions.

Answer: Outdoor testing of WD-ER unit for practical applications were performed while connected to a battery to process the electricity generated by the PV cell. The generated electricity was supplied to a battery with a low voltage, and did not increase the temperature of the cell. However, the main purpose of this study is thermal management of the PV cell, so this content was not included in the manuscript. The I-V curve in terms of temperature variation was evaluated and the performance/behavior of PV was basically verified.

The testing details for processing electricity was reflected on Practical applications of METHODS as follows;

Practical applications: A PV cell (a square size of 14.5 cm) with WD-ER cooling unit is illustrated in Figure S5. The urethane waterproofing coating was applied to protect the power terminal unit on the back of the PV cell, and the thickness was approximately 0.2 mm. Further, 20 wt% zeolite 13X and 1 wt% polyvinyl alcohol (PVA) were dispersed in ethanol. After 8 h at 50 °C in a convection oven, zeolite 13X was coated on the surface of the PV cell, and it was repeated until the 13X coating layer had a thickness of 2 mm. Zeolite 13X was fully saturated by exposure to a moist environment. Subsequently, ammonium nitrate was coated by the same method, and the total mass was adjusted to an optimal composition. Because moisture was generated when heat was absorbed, it was sealed with an acrylic case. The PV cell with the WD-ER cooling unit was placed in a suitable position for power generation, and the cooling performance was estimated during a day. The experiment was conducted from 10 am to 5 pm with an average ambient temperature of 30.5 °C and a maximum temperature of 35.4 °C. The temperature variation was collected at 10 s intervals utilizing a midi LOGGER GL240 manufactured by GRAPHTEC. The current-voltage characteristics according to the temperature of the PV cell were measured with a PROVA-200A manufactured by TES. The experiments were performed while the PV cell was connected to a battery to process the generated electricity.

9. In Fig. 8a, it is obvious that a hot spot is found on the center of the cell and the monitored temperature is overestimated, which directly affect the reported temperature reduction in conclusion. This phenomenon may be contributed by the setting of thermocouples, please check the experimental setting.

Answer: Thank you for your good comment. It is correct that the temperature appeared high as the thermocouple was inserted in the center. If the temperature is measured at the bottom of the PV cell, the actual temperature of cell surface may be distorted, and therefore the thermocouple

was inserted in the center for precise experiments. Nevertheless, repeated experiments demonstrated sufficient reproducibility.

Thermal imaging camera measurement results are attached as below. As can be seen in the results, the temperature gradients according to the position is significant in the case of the natural air cooling while the temperature distribution is uniform due to the high cooling performance in the case of the WD-ER unit.

Natural air cooling:

Day 1

Day 2

Day 3

Day 4

WD-ER unit:

Day 1

Day 2

Day 3

Day 4

REVIEWER COMMENTS

Reviewer #1 (Remarks to the Author):

The authors have addressed some of my concerns regarding the novelty statement. However, the overall writing and organization of this manuscript still need a lot of work. There are too many figures in the main text. Many figure data in the main text are just supporting characterization data. They should be reorganized, merged, or moved to the supporting information.

Reviewer #2 (Remarks to the Author):

See attachment

Self-healing passive cooling utilizing endothermic reaction of $\text{NH}_4\text{NO}_3/\text{H}_2\text{O}$ driven by moisture sorption for photovoltaic cell

Decision: Accept

Review Comments

The authors have completed a sincere and major revision of the manuscript to appropriately address all raised issues. The manuscript merits publication considering the reported performance enhancement compared to other passive and active cooling technologies.

Reviewer #3 (Remarks to the Author):

Comments:

The authors have made modifications to the manuscript to improve the quality of the work. I still have some questions and suggestions. Please find my comments below.

1. A cooling comparison has been conducted in Table 1. Radiative cooling has been selected as one of the important cooling methods. Ref. 49 is not appropriate in Table 1 since sub-ambient radiative cooling is the topic of the ref. 49. Radiative cooling of solar cells is above-ambient radiative cooling. In addition, another important cooling method that combines radiative cooling and spectrally solar harvesting is not described, such as [10.1021/acsp Photonics.7b00089](https://doi.org/10.1021/acsp Photonics.7b00089), [10.1016/j.solmat.2018.01.023](https://doi.org/10.1016/j.solmat.2018.01.023)
2. In revised Figures 1(c) and (d), I suggest that using the heat flux direction rather than the cooling effect direction is better for understanding.
3. In outdoor testing, a continuous temperature curve (at least 2 days) should be given for solar cells with/without the cooling unit. If so, the cooling effect of solar cells is visualized for general readers.
4. In Figure 7(a), although explanations and supplementary testing are provided to demonstrate that the current setting of attaching the thermocouple to the front side is reasonable, I still question this measurement. First, this setting will create shadows for solar cells, which makes different effects on the PV process. Second, this setting changes the temperature distribution of the temperature field, which affects the measured temperature value. For example, the thermal image shows that the temperature of the cell center (with a thermocouple attached) in the natural air cooling case is higher than that of the other position. If this measured temperature is used to evaluate the cooling effect, the temperature reduction of the cell will be overestimated.

Title: Self-recovering passive cooling utilizing endothermic reaction of $\text{NH}_4\text{NO}_3/\text{H}_2\text{O}$ driven by water sorption for photovoltaic cell

Reviewers' comments

Reviewer #1

The authors have addressed some of my concerns regarding the novelty statement. However, the overall writing and organization of this manuscript still need a lot of work. There are too many figures in the main text. Many figure data in the main text are just supporting characterization data. They should be reorganized, merged, or moved to the supporting information.

Answer: Thank you very much for evaluating my manuscript with interest and allowing us to make a lot of progress based on your effusive comments and dedications. The organization has been improved by referring to the issues you mentioned and *Nature Communications* formatting instructions.

Table of contents (Before):

INTRODUCTION

RESULTS AND DISCUSSION

Working principle of self-recovering passive cooling

Porous materials for latent cooling

Solvent pairs for endothermic reaction cooling

Lab-scale heat dissipation performance using WD-ER

Practical applications of WD-ER for photovoltaic cell cooling

CONCLUSIONS

METHODS

Materials preparation

Characterization

Theory of endothermic reaction cooling

Lab-scale heat dissipation experiments

Practical applications

Data availability

Acknowledgement

References

Table of contents (After):

Introduction

Results

Working principle of self-recovering passive cooling

Porous materials for latent cooling

Solvent pairs for endothermic reaction cooling

Lab-scale heat dissipation performance of WD-ER

Practical applications of WD-ER for photovoltaic cell cooling

Discussion

Methods

Materials preparation

Characterization

Theory of endothermic reaction cooling

Lab-scale heat dissipation experiments

Practical applications

Data availability

References

Acknowledgements

Author contribution

Competing interests

“Abstract” was revised to be concise within 150 words and contain the key contributions of the manuscript.

Before (189 words):

Power efficiency of photovoltaic (PV) cell is significantly affected by the cell temperature. Here, a self-recovering passive cooling unit comprising inexpensive materials is developed. The water-saturated zeolite 13X is coated on the back side of the PV cell, and ammonium nitrate is dispersed as a layer to form a thin film. When thermal radiation is supplied, water is desorbed from zeolite 13X (primary latent cooling), and dissolves ammonium nitrate to induce secondary endothermic reaction cooling. The cooling unit works on the basis that the water sorption performance of microporous materials is inversely proportional to the cell temperature, and the solubility of endothermic reaction pairs increases proportionally with temperature. It is a reversible process in which ammonium nitrate is crystallized when the temperature is low at night, and water is adsorbed back to zeolite 13X. The average temperature of the PV cell can be reduced by 15.1 °C during the day, and the cooling energy density reaches 2,876 kJ/kg with average cooling power of 403 W/m², indicating a significant improvement compared to the reported passive cooling methods. These results suggest that the highly efficient solar energy harvesting could be achieved.

After (150 words):

Power efficiency of photovoltaic (PV) cell is significantly affected by the cell temperature. Here, **a self-recovering passive cooling unit is developed**. The water-saturated zeolite 13X is coated on the back side of PV cell, and ammonium nitrate is dispersed as a layer to form a thin film. When **heat** is supplied, water is desorbed from zeolite 13X (**latent cooling**), and dissolves ammonium nitrate to induce **endothermic reaction cooling**. **It is a reversible process that recovers itself at night**. The unit works on the basis that the water sorption performance of **porous materials** is inversely proportional to temperature, and the solubility of endothermic reaction pairs increases proportionally with temperature. The average temperature of PV cell can be reduced by 15.1 °C, and the cooling energy density reaches 2,876 kJ/kg with average cooling power of 403 W/m². **We show that highly efficient passive cooling comprising inexpensive materials for PV cell could be achieved.**

“Conclusions” were changed to “Discussion”. In addition, the key contributions and future research of this work were covered.

Before:

CONCLUSIONS

A novel type of self-recovering passive cooling unit comprising inexpensive materials without maintenance problems has been developed for PV cell applications. It is defined as a water desorption-driven endothermic reaction (WD-ER) cooling unit, and presents the best heat dissipation performance among the methods reported in the literature.

The WD-ER cooling unit comprises a water-saturated zeolite 13X and an ammonium nitrate crystal layer. The water sorption performance of porous materials is inversely proportional to temperature, and the solubility of ammonium nitrate in the solvent increases with temperature. When thermal energy is supplied, the temperature of zeolite 13X increases, heat is absorbed for water desorption in the pores (primarily latent cooling; 346.6 kJ/kg-carrier), and liquid water dissolves ammonium nitrate crystals (secondary endothermic reaction cooling; 1,386 kJ/kg-carrier). By the chain reaction of water desorption and solute dissolution, the overall cooling energy density of the WD-ER unit reaches 2,876 kJ/kg, and it can be optimized by considering the required cooling energy density. When the temperature is lowered at night, it is self-recovered, such that the ammonium nitrate is crystallized and water is adsorbed into zeolite 13X. The WD-ER cooling unit is then utilized continuously. The average effective heat transfer coefficient is 64.1 W/m²·K, which is higher than that of forced air cooling (12.1 W/m²·K).

For industrial applications of WD-ER cooling unit with PV cell composed of monocrystalline silicon, the temperature of the cell can be reduced by 15.1 °C during the day, and this improves the power efficiency by 10%. In the temperature range of 20–45 °C, the latent cooling effect is dominant, and the supplied heat is utilized for the desorption of water from zeolite 13X (cooling power: 380 W/m²). In addition, the combined cooling mode of water desorption and dissolution of the solute is induced as the ammonium nitrate layer collapses above 45 °C with the cooling power: 431 W/m².

After:

Discussion

Inspired by recent studies on passive cooling strategy for PV cell, which improves power generation efficiency and durability of the cell, a novel type of self-recovering passive cooling unit comprising inexpensive materials without maintenance problems is developed. It is defined as a water desorption-driven endothermic reaction (WD-ER) cooling unit. The WD-ER cooling unit comprises a water-saturated zeolite 13X and an ammonium nitrate crystal layer to form a thin film. The water sorption performance of porous materials is inversely proportional to temperature, and the solubility of ammonium nitrate in the solvent increases with temperature. When thermal energy is supplied, the temperature of zeolite 13X increases, heat is absorbed for water desorption in the pores (primarily latent cooling; 346.6 kJ/kg-carrier), and liquid water dissolves ammonium nitrate crystals (secondary endothermic reaction cooling; 1,386 kJ/kg-carrier). By the chain reaction of water desorption and solute dissolution, the overall cooling energy density of the WD-ER unit reaches 2,876 kJ/kg, and it can be optimized by considering the required cooling energy density. When the temperature is lowered at night, it is self-recovered, such that the ammonium nitrate is crystallized and water is adsorbed into zeolite 13X.

For industrial applications of WD-ER cooling unit with PV cell composed of monocrystalline silicon, the temperature of the cell can be reduced by 15.1 °C during the day. In the temperature range of 20–45 °C, the latent cooling effect is dominant, and the supplied heat is utilized for the desorption of water from zeolite 13X (cooling power: 380 W/m²). In addition, the combined cooling mode of water desorption and dissolution of the solute is induced as the ammonium nitrate layer collapses above 45 °C with the cooling power of 431

W/m². The average cooling power density is uniform at 375–419 W/m² during cyclic cooling experiments when outdoor temperature ranges 30.5–31.7 °C. The WD-ER unit outperforms the cooling methods reported in the literature. Note that, as only different types of zeolites are analyzed for WD-ER cooling unit in this study, further improvement would be expected by utilizing advanced approaches including metal-organic frameworks.

Many figures in the main text was moved to Supporting Information, and the other figures were reorganized/merged to focus on direct proof of the main claim.

Before:

Figure 1. Schematic diagram of self-recovering passive cooling unit utilizing the chain reaction of water desorption (primary latent cooling) and dissolution of ammonium nitrate in water (secondary endothermic reaction cooling) for PV cell applications; (a) Passive cooling unit integrated with PV cell; (b) Cooling power during electricity generation in response to solar radiation; (c) Cooling principle considering internal composition during the day; (d) Self-recovering process during the night.

Figure 2. Water sorption performance of porous materials; (a) Zeolite 13X; (b) Zeolite 5A; (c) Zeolite 3A; (d) Zeolite Y; (e) SAPO-34; (f) SSZ-13.

Figure 3. Latent cooling behavior of porous materials; (a) Reaction heat versus water loading; (b) Latent cooling energy density considering water sorption capacity and heat of reaction.

Figure 4. Theoretical analysis of endothermic reaction pairs for WD-ER cooling; (a) Ideal solubility and heat of reaction; (b) Endothermic reaction cooling energy density.

Figure 5. Heat dissipation performance in terms of cooling methods; (a) Natural air cooling; (b) Forced air cooling; (c) Latent cooling; (d) Endothermic reaction cooling; (e) WD-ER cooling; (f) Repeated WD-ER cooling.

Figure 6. Heat dissipation performance of latent cooling in terms of zeolite film thickness

Figure 7. Application of WD-ER unit for PV power generation; (a) Schematic of PV cell with WD-ER unit; (b) Temperature variation of PV cell during a day; (c) Net cooling power density of WD-ER unit without natural convection.

Figure 8. Practical working performance of WD-ER unit; (a) Cooling power density and average temperature variation during the day; (b) Current-voltage characteristics in terms of temperature.

Figure S1. Characterization of porous materials; (a) N₂ adsorption curve; (b) XRD pattern; (c) Micropore size distribution; (d) Mesopore size distribution.

Figure S2. FE-SEM images for porous materials; (a) Zeolite 13X; (b) Zeolite 5A; (c) Zeolite 3A; (d) Zeolite Y; (e) SAPO-34; (f) SSZ-13

Figure S3. Visualization analysis during heat dissipation process; (a) Schematic of thermal imaging experiment; (b) Latent cooling using zeolite 13X/H₂O; (c) Endothermic reaction cooling using NH₄NO₃/H₂O

Figure S4. Experimental device for evaluation of heat dissipation performances

Figure S5. Manufacturing method of PV cell with WD-ER cooling unit; (a) PV cell composed of monocrystalline silicon; (b) Waterproofing coating to protect power terminal unit; (c) water-saturated zeolite 13X coating for latent cooling; (d) Ammonium nitrate coating for endothermic reaction cooling; (e) Sealing unit for safe operation.

After:

Fig. 1. Overview of the passive cooling unit. a Schematic representation of self-recovering passive cooling unit utilizing the chain reaction of water desorption (primary latent cooling)

and dissolution of ammonium nitrate in water (secondary endothermic reaction cooling) integrated with photovoltaic (PV) cell. **b** Cooling power during electricity generation in response to solar radiation. **c** Cooling principle considering internal composition during the day. **d** Self-recovering process during the night.

Fig. 2. Latent cooling behavior of porous materials. a Water sorption performance of zeolite 13X. **b** Latent cooling energy density considering water sorption capacity and heat of reaction.

Fig. 3. Theoretical analysis of endothermic reaction pairs for water desorption-driven endothermic reaction (WD-ER) cooling. a Ideal solubility and heat of reaction. **b** Endothermic reaction cooling energy density.

Fig. 4. Heat dissipation performance in terms of cooling methods. a Latent cooling. **b** Endothermic reaction cooling. **c.** Water desorption-driven endothermic reaction (WD-ER) cooling. **d** Repeated WD-ER cooling.

Fig. 5. Heat dissipation performance of latent cooling in terms of zeolite film thickness

Fig. 6. Practical application of water desorption-driven endothermic reaction (WD-ER) cooling unit for photovoltaic (PV) cell. a Images of PV cell with WD-ER unit. **b** Temperature variation of PV cell during a day. **c** Net cooling power density of WD-ER unit without natural convection. **d** Practical cyclic working performance of WD-ER unit.

Fig. S1. Characterization of porous materials. a N₂ adsorption curve. **b** XRD pattern. **c** Micropore size distribution. **d** Mesopore size distribution.

Fig. S2. FE-SEM images for porous materials. a Zeolite 13X. **b** Zeolite 5A. **c** Zeolite 3A. **d** Zeolite Y. **e** SAPO-34. **f** SSZ-13

Fig. S3. Water sorption performance of porous materials. a Zeolite 5A. **b** Zeolite 3A. **c** Zeolite Y. **d** SAPO-34. **e** SSZ-13.

Fig. S4. Reaction heat versus water loading of porous materials

Fig. S5. Visualization analysis during heat dissipation process. a Schematic of thermal imaging experiment. **b** Latent cooling using zeolite 13X/H₂O. **c** Endothermic reaction cooling using NH₄NO₃/H₂O

Fig. S6. Experimental device for evaluation of heat dissipation performances

Fig. S7. Heat dissipation performance in terms of cooling methods. a Natural air cooling. **b** Forced air cooling.

Fig. S8. Current-voltage characteristics in terms of temperature.

Fig. S9. Manufacturing method of PV cell with WD-ER cooling unit. a PV cell composed of monocrystalline silicon. **b** Waterproofing coating to protect power terminal unit. **c** Water-saturated zeolite 13X coating for latent cooling. **d** Ammonium nitrate coating for endothermic reaction cooling. **e** Sealing unit for safe operation.

Reviewer #2

Decision: Accept

The authors have completed a sincere and major revision of the manuscript to appropriately address all raised issues. The manuscript merits publication considering the reported performance enhancement compared to other passive and active cooling technologies.

Answer: I would like to extend my sincerest gratitude to the reviewer #2 for his/her valuable feedback and constructive criticism during the revision process. Your input has greatly

improved the quality. Thank you for your time/effort in reviewing our work.

Reviewer #3

The authors have made modifications to the manuscript to improve the quality of the work. I still have some questions and suggestions. Please find my comments below.

Answer: I want to express my gratitude for the constructive feedback you provided during the previous revision process, which significantly improve the quality of the manuscript. I am pleased to inform you that I have carefully incorporated your present comments and I hope you find our revised manuscript to be satisfactory.

1. A cooling comparison has been conducted in Table 1. Radiative cooling has been selected as one of the important cooling methods. Ref. 49 is not appropriate in Table 1 since sub-ambient radiative cooling is the topic of the ref. 49. Radiative cooling of solar cells is above-ambient radiative cooling. In addition, another important cooling method that combines radiative cooling and spectrally solar harvesting is not described, such as 10.1021/acsp Photonics.7b00089, 10.1016/j.solmat.2018.01.023

Answer: Raman et al. (Ref. 49. Passive radiative cooling below ambient air temperature under direct sunlight, Nature 2014) demonstrated radiative cooling to nearly 5 °C below the ambient air temperature under direct sunlight utilizing a thermal photonic approach of seven layers of HfO₂ and SiO₂. It was excluded from Table 1 because it was not appropriate, as it operated at the below ambient air temperature (Previously, it was put in Table 1 based on that the mechanism is similar to that of above-ambient radiative cooling for PV cell).

In addition, Table 1 was revised to describe another important cooling methods that combine radiative cooling and spectrally solar harvesting based on the suggested references as follows;

Before:

Table 1. Comparison of WD-ER cooling unit with passive cooling methods

Methods		Density (kJ/kg)	Conditions and remarks	References
WD-ER	13X/H ₂ O/NH ₄ NO ₃	2,876	Operating range: 20-70 °C Cooling power: 403 W/m ² Temperature drop: 15.1 °C	
Phase change materials	RT55	170	Melting: 51-57 °C	40
	RT28HC	220	Melting: 28 °C	41
	C-PCM	199	Melting: 21.8 °C	42
	OM47	196	Melting: 48 °C	43
	MF-PW30	139.8	Melting: 56.8 °C, crystallization: 45.1 °C	44
	PCMB	141.7	Steady range: 40.4-84.9 °C	45
	GO-EHS/PAAAM	200.3	Melting: 23 °C	46

	Eicosane	237.4	Melting: 36.5 °C	47
	Tricosane	269.2	Melting: 42-48 °C	48
	MIL-101(Cr)	1,950	Operating range: 20-70 °C	31
	PAN-CNT-CaCl ₂	-	Operating range: 20-70 °C Cooling power: 295 W/m ² Temperature drop: 10.0 °C	18
Radiative cooling	HfO ₂ and SiO ₂	-	Operating range: 22 °C Cooling power: 40.1 W/m ²	49
	Cooling assembly system	-	Operating range: 74 °C Cooling power: 310 W/m ² Temperature drop: 36.6 °C	50
	Silica pyramid	-	Operating range: 20-70 °C Temperature drop: 18.3 °C	51

After:

Table 1. Comparison of WD-ER cooling unit with passive cooling methods

Methods		Density (kJ/kg)	Conditions and remarks	Ref .
WD-ER	13X/H ₂ O/NH ₄ NO ₃	2,876	Operating range: 20-70 °C Cooling power: 403 W/m ² Temperature drop: 15.1 °C	
Phase change materials	RT55	170	Melting: 51-57 °C	40
	RT28HC	220	Melting: 28 °C	41
	C-PCM	199	Melting: 21.8 °C	42
	OM47	196	Melting: 48 °C	43
	MF-PW30	139.8	Melting: 56.8 °C, crystallization: 45.1 °C	44
	PCMB	141.7	Steady range: 40.4-84.9 °C	45
	GO-HS/PAAAM	200.3	Melting: 23 °C	46
	Eicosane	237.4	Melting: 36.5 °C	47
	Tricosane	269.2	Melting: 42-48 °C	48
	MIL-101(Cr)	1,950	Operating range: 20-70 °C	31
	PAN-CNT-CaCl ₂	-	Operating range: 20-70 °C Cooling power: 295 W/m ² Temperature drop: 10.0 °C	18
Radiative cooling	Cooling assembly system	-	Operating range: 74 °C Cooling power: 310 W/m ² Temperature drop: 36.6 °C	49
	Silica pyramid	-	Operating range: 20-70 °C Temperature drop: 18.3 °C	50
	Multilayer photonic film (Al ₂ O ₃ /SiN/TiO ₂ /SiO ₂)	-	Operating range: 40-70 °C Cooling power: 149 W/m ² Temperature drop: 5.7 °C	19
	Multilayer stack (SiO ₂ /TiO ₂ /MgF ₂)	-	Operating range: 10-30 °C Cooling power: 29.5 W/m ²	51

- 19 Li, W., Shi, Y., Chen, K., Zhu, L. & Fan, S. A comprehensive photonic approach for solar cell cooling. *ACS Photonics* **4**, 774-782 (2017).
- 51 Zhao, B., Hu, M., Ao, X., Xuan, Q. & Pei, G. Comprehensive photonic approach for diurnal photovoltaic and nocturnal radiative cooling. *Solar Energy Materials and Solar Cells* **178**, 266-272 (2018).

2. In revised Figures 1(c) and (d), I suggest that using the heat flux direction rather than the cooling effect direction is better for understanding.

Answer: Figures 1c and 1d were revised using the heat flux direction rather than the cooling effect direction for better understanding as follows;

Before:

After:

3. In outdoor testing, a continuous temperature curve (at least 2 days) should be given for solar cells with/without the cooling unit. If so, the cooling effect of solar cells is visualized for general readers.

Answer: Thank you for your good comment. A continuous temperature curve for 2 days was added in Figure 6.

Before:

Figure 7. Application of WD-ER unit for PV power generation; (a) Schematic of PV cell with WD-ER unit; (b) Temperature variation of PV cell during a day; (c) Net cooling power density of WD-ER unit without natural convection.

After:

a

Fig. 6. Practical application of water desorption-driven endothermic reaction (WD-ER) cooling unit for PV cell. a Images of PV cell with WD-ER unit. **b** Temperature variation of PV cell during a day. **c** Net cooling power density of WD-ER unit without natural convection. **d** Practical cyclic working performance of WD-ER unit.

4. In Figure 7(a), although explanations and supplementary testing are provided to demonstrate that the current setting of attaching the thermocouple to the front side is reasonable, I still question this measurement. First, this setting will create shadows for solar cells, which makes different effects on the PV process. Second, this setting changes the temperature distribution of the temperature field, which affects the measured temperature value. For example, the thermal image shows that the temperature of the cell center (with a thermocouple attached) in the natural air cooling case is higher than that of the other position. If this measured temperature is used to evaluate the cooling effect, the temperature reduction of the cell will be overestimated.

Answer: Thanks for your important point. The current setting of attaching the thermocouple to the front side was further analyzed as below;

The area where the tape and thermocouple are attached to the PV cell is shown in the figure below.

The surface area where the tape is attached is only 8% of the total area of PV cell, and it does not have a significant effect.

Thermal resistance for temperature analysis is shown in the figure below: (a) represents thermal resistance for PV cell and (b) represents thermal resistance for PV cell with the tape. As the PV cell is very thin, the temperature gradient in the length direction can be ignored (Biot number is much smaller than 0.1). In addition, it is reasonable to assume that the thermocouple and PV cell are in thermal equilibrium because the thermocouple is in contact with the PV cell.

The heat supplied by the sun is 1,000 W/m². Monocrystalline silicon has a solar absorptivity of 0.82, and 820 W/m² is absorbed. On the other hand, by assuming that the tape is black body (absorptivity = 1), it absorbs 1,000 W/m².

As 8% of the total area (tape) absorbs about 18% more heat, it roughly affects the entire temperature field by only 1.44%.

The temperature of PV cell under the tape can also be obtained by Fourier's law of conduction below, where k is thermal conductivity of the tape (0.05 W/m·K), t is thickness (0.25 mm), T_{it} is temperature of the tape, and T_{PV} is temperature of PV cell under the tape.

$$1,000 \text{ W/m}^2 \text{ (absorbed heat)} = k \cdot \frac{T_{it} - T_{PV,it}}{t}$$

T_{it} can be obtained from thermal imaging data as follows. Then, $T_{PV,it}$ is 52.8 °C which is very close to the measured value of 53.1 °C.

Also, we confirmed that the thermal imaging data and the thermocouple data collected at the same time are very similar within ± 0.5 °C.

We consider that readers may raise similar questions if we omit this important discussion. Therefore, it was reflected in the paper as follows.

Practical applications of WD-ER for photovoltaic cell cooling

The WD-ER cooling unit is applied to a PV cell comprising monocrystalline silicon. As illustrated in **Figure 6a**, urethane waterproofing on the back of the PV cell protects the power terminal unit from moisture penetration. Zeolite 13X particles are coated on the PV cell with a thickness of 2 mm, and a negligible amount of polyvinyl acetate is included. After the coated zeolite 13X is saturated with water, the NH_4NO_3 crystal layer is added to configure the WD-ER cooling unit (The configuration of PV cell with WD-ER cooling unit for industrial applications is same as that of lab-scale heat dissipation experiment in **Figure 4**). PV cells with natural air cooling and WD-ER cooling units were installed in adjacent positions to observe the temperature variation. Based on the images taken with a thermal imaging camera at 2 pm, the surface temperature of the PV cell with the WD-ER cooling unit is 39–43°C, which is significantly lower than that of natural air cooling (51–56 °C) as shown in **Figure 6a**. **Note that although the tape for attaching the thermocouple has a higher thermal absorptivity than monocrystalline silicon, the effect on the temperature field of PV cell can be neglected in that it accounts for only 8% of the total area. When the temperature of PV cell with the tape is analyzed by Fourier's law, the temperature deviation from other PV cell parts is within 0.3 °C.**

REVIEWERS' COMMENTS

Reviewer #3 (Remarks to the Author):

My concerns have been addressed appropriately.